# Application of a Hygroscopicity Tandem Differential Mobility Analyzer for characterizing PM Emissions in exhaust plumes from an Aircraft Engine burning Conventional and Alternative fuels

Max B. Trueblood[1], Prem Lobo[1,a], Donald E. Hagen[1], Steven C. Achterberg[1], Wenyan Liu[2], Philip D. Whitefield[1]

[1]Center of Excellence for Aerospace Particulate Emissions Reduction Research, Missouri University of Science and Technology, Rolla, Missouri, USA 65409

[2]Center for Research in Energy and Environment, Missouri University of Science and Technology, Rolla, Missouri, USA 65409

[a] Now at: Metrology Research Centre, National Research Council Canada, Ottawa, Ontario, Canada K1A 0R6

*Correspondence to: Max B. Trueblood (trueblud@mst.edu)*

**Abstract.** In the last several decades, significant efforts have been directed toward better understanding the gaseous and Particulate Matter (PM) emissions from aircraft gas turbine engines. However, limited information is available on the hygroscopic properties of aircraft engine PM emissions which play an important role in the water absorption, airborne lifetime, obscuring effect, and detrimental health effects of these particles. This paper reports the description and detailed lab-based performance evaluation of a robust Hygroscopicity-Tandem Differential Mobility Analyzer (H-TDMA), in terms of hygroscopic properties such as growth factor (GF) and the hygroscopicity parameter (κ). The H-TDMA system was subsequently deployed during the Alternative Aviation Fuel EXperiment (AAFEX) II field campaign to measure the hygroscopic properties of aircraft engine PM emissions in the exhaust plumes from a CFM56-2C1 engine burning several types of fuels. The fuels used were conventional JP-8, tallow-based hydro-processed esters and fatty acids (HEFA), Fischer-Tropsch, a blend of HEFA and JP-8, and Fischer-Tropsch doped with Tetrahydrothiophene (an organosulfur compound). It was observed that GF and κ increased with fuel sulfur content and engine thrust condition, and decreased with increasing dry particle diameter. The highest GF and κ values were found in the smallest particles, typically those with diameters of 10 nm.

## 1 Introduction

The increase in aviation related activities has led to concern about the emissions from aircraft operations and their impact on local air quality (Unal et al., 2005; Woody et al., 2011), global climate (Lee et al., 2009; Brasseur et al., 2016), and public health (Levy et al., 2012; Brunelle-Yeung et al., 2014). The primary products of conventional jet fuel combustion in an aircraft engine are NOx, UHC, CO, SOx, $CO_2$, $H_2O$, and soot aerosol or soot particulate matter (PM). As the aircraft engine exhaust plume expands, mixes with ambient air, and cools, volatile species present in the gas phase at the engine exit plane undergo

gas-to-particle conversion, and begin to condense onto existing soot particles and form new particles (Onasch et al., 2009; Lobo et al., 2012; Timko et al., 2013). The black carbon component of the PM is referred to as non-volatile particulate matter (nvPM), while the volatile component consists of sulfates, nitrates, and organic compounds (Onasch et al., 2009). The composition of the volatile PM in the expanding aircraft engine exhaust plume varies greatly, and depends on a number of factors such as fuel composition, ambient meteorological conditions, and plume age (Lobo et al., 2007; Lobo et al., 2012; Timko et al., 2013; Lobo et al., 2015a).

The commercial aviation sector has been focussed on developing and implementing sustainable alternative jet fuels for use by airlines to diversify fuel supplies and mitigate the impacts of aircraft engine emissions. The American Society for Testing and Materials (ASTM) and other fuels specification bodies have established a standard specification for the manufacture of aviation turbine fuel consisting of conventional and synthetic blending components under ASTM D7566 (ASTM, 2016). The pure alternative fuels have low to negligible amounts of aromatic, naphthalenes, and sulfur content when compared to conventional jet fuel. Studies have shown that nvPM and sulfur oxide emissions are dramatically reduced during alternative fuel combustion in aircraft engines (Timko et al., 2010; Lobo et al., 2011; Beyersdorf et al., 2014; Moore et al., 2015; Lobo et al., 2015b; Lobo et al., 2016). The nvPM at the engine exit plane is hydrophobic, but as the nvPM evolves in the expanding plume, its aging results in enhanced hydrophilicity (Weingartner et. al., 1997; Zhang et. al., 2008).

Investigation of atmospheric pollution, and in particular atmospheric visibility, has shown that aerosol optical properties are affected by size, composition, and hygroscopic growth of particles (Tang et al., 1981; Horvath, 1995; Kim et al., 2006; Meier et al., 2009). In urban environments, emissions from vehicles including soot, sulfates, and nitrates have been found to be the main contributors to visibility degradation (Ferron et al., 2005; Kim et al., 2006).

Hygroscopicity-Tandem Differential Mobility Analysis (H-TDMA) systems have been widely used to measure the hygroscopic growth properties of PM in the sub-saturated regime in different environments (Massling et al., 2007; Swietlicki et al., 2008; Park et al., 2009b; Wu et al., 2013). H-TDMA measurements of PM emissions from jet engine combustors (Gysel et al., 2003; Popovicheva et al., 2008) have also been performed. However, the application of an H-TDMA system to measure the hygroscopic properties of PM emissions measured in evolving aircraft engine exhaust plumes from the combustion of different fuels has not been previously performed.

For field measurements, where ambient temperature and humidity cannot be controlled, the H-TDMA system must be fairly rugged, stable, and versatile. The Missouri University of Science and Technology (MST) has developed a H-TDMA system to quantify the hygroscopic properties of PM emitted from aircraft engines. The H-TDMA system was automated to operate such that it could determine the hygroscopic properties for an aerosol in approximately 45 seconds. This is critical when conducting aircraft engine emission tests which can be quite expensive, and where the expanding exhaust plumes are subject to perturbations in wind speed and wind direction. This paper reports the results of lab-based experiments to evaluate the performance of the MST H-TDMA system, and in-field measurements of PM emissions in exhaust plumes from the combustion of conventional and alternative fuels in a CFM56-2C1 engine during the Alternative Aviation Fuels EXperiment (AAFEX) II field campaign.

## 2 Experimental Method

The MST H-TDMA system consists of two differential mobility analyzers (DMAs), a humidifier (HUM), and a condensation particle counter (CPC), similar to other systems (McMurry et al., 1989).  Fig 1 presents the schematic of the MST H-TDMA system. The polydisperse aerosol was first pre-conditioned by passing it through an ice bath (IB-0) to remove excess water vapor as much as reasonably possible and return it to room temperature with a saturation ratio of ~ 0.15. The aerosol was then brought to charge equilibrium by passing it through a bipolar charger (BC), which can contain 500 to 2,000 µCi of Polonium-210 prior to entering the first DMA (DMA1). The DMAs used in the H-TDMA system were custom designed and have been used in previous studies to classify aerosols based on electrical mobility (Schmid, 2000). The DMAs were of cylindrical geometry and had the following dimensions: effective inner length of 72.77 cm, and a sample flow annulus with an inner diameter of 5.07 cm and an outer diameter of 8.88 cm. The polydisperse aerosol flow rate ($Q_p$) was set to 3 L min$^{-1}$ and the sheath flow rate ($Q_s$) was adjusted to 15 L min$^{-1}$ using mass flow meters (Aalborg Instruments GFM 371) which were calibrated periodically. In DMA1, the polydispersed aerosol was classified by size, and monodisperse particles with a "dry" size ($X_d$) were selected.  The excess flow in the DMA was recirculated as $Q_{s1}$, after passing through a second ice bath (IB-1) and a HEPA filter to further ensure that the sample remained dry and had not prematurely deliquesced to a solution droplet. DMA1 was set at a fixed voltage permitting the selection of a monodisperse aerosol. The monodisperse sample flow ($Q_{m1}$) out of DMA1 entered the humidifier (HUM) section of the H-TDMA system, where it is referred to as the polydisperse flow, $Q_{p2}$. The HUM brought the aerosol sample to a controlled, precisely known saturation ratio (SR), typically 0.91  0.99, which caused the particles to deliquesce to a new equilibrium "wet" diameter ($X_w$).  Valves V2 and V3 were used to direct the aerosol flow $Q_{p2}$ to either pass through HUM (wet mode) or to bypass it (dry mode). Valves V4 and V5 were used to achieve the same function for the sheath air flow ($Q_{s2}$). The third ice bath (IB-2) in the $Q_{s2}$ loop removed the water vapor from $Q_{s2}$ and minimized any unwanted vapors co-emitted from the combustion process.  The second DMA (DMA2) in conjunction with a CPC (TSI 3022) measured $X_w$. The MST H-TDMA system was designed to provide only one SR condition, and to hold that value regardless of variations in ambient temperature and humidity or sampling duration. The water bath that encased HUM/DMA2 was maintained at a fixed temperature by a refrigerated water re-circulator that controlled the water temperature around the HUM/DMA2 to 16. ± 0.1 °C.  This water passed alongside the $Q_{p2}$ and $Q_{s2}$ lines (not shown in figure). Thus the dew point achieved in HUM was well below room temperature. The water flow rate through the water bath surrounding HUM/DMA2 was approximately 5 L min$^{-1}$.

The SR values in flows $Q_{p2}$ and $Q_{s2}$ were brought to near unity at 16 °C by passing the aerosol through stainless steel tubes lined with wet cloth. The flow $Q_{p2}$ passed through 4 such tubes (11 mm ID x 762 mm L), thus having a total length of 3048 mm and a residence time of 5.8 s.  The flow $Q_{s2}$ passed through 8 similar tubes, thus having a total length of 6096 mm and a residence time of 2.3 s.  Theoretical studies have shown that the lengths of wet walled tubing should be sufficient to bring the $Q_{p2}$ and $Q_{s2}$ to very near SR=1 (Fitzgerald et al., 1980).  Just before entering DMA2, the SR of $Q_{s2}$ was measured by a dew

point hygrometer (DPH) (Vaisala HMP247). The flow $Q_d$ in parallel with the CPC, reduced the lag time (LT2) between when a voltage was imposed on DMA2 and when particles selected by that voltage reached the CPC.

During routine operation, to maximize the data acquisition frequency, the H-TDMA system was computer controlled by a LabVIEW program (LV). When the program was initiated, it (1) set the desired voltage (HV1) in DMA1 causing it to deliver dry particles of diameter $X_d$, (2) waited long enough for this monodisperse aerosol to travel from the outlet of DMA1 through the HUM and into DMA2, (3) set the high voltage in DMA2 (HV2) to some fraction of that in DMA1 (typically 0.1 x HV1), and (4) caused HV2 to step through 104 increments such that the final value was a multiple of HV1 (typically 10 x HV1). During the stepwise voltage increase of HV2 (the logarithm of the voltage was linear with time.), LV recorded (at 1 Hz) values of HV1, HV2, $Q_{s1}$, $Q_{s2}$, $Q_d$, P1, P2, SR, CPC concentration and elapsed time (dt). The operator provided the general region (in time) where the peak in CPC readings occurred as input, and LV fitted a quadratic function to the CPC concentration time series. The quadratic function was differentiated and the value of dt at the maximum was obtained ($dt_{max}$). Based on calibrations performed previously, LV computed the lag time (LT2) between when a certain diameter of droplet was selected by DMA2 and when it arrived at the CPC. This lag time has been found to be a function of $Q_{s2}$ and $Q_{p2}$. LV found the value of the high voltage on the central rod of DMA2 at that time. It then computed the wet diameter ($X_w$) of the solution droplet (using the operating equation of the DMA2), and finally computed the hygroscopic peroperties. LV was developed such that the hygroscopic properties could be determined on more than one $X_d$. LV changes the particle diameter produced by DMA1 before the end of the voltage sweep on DMA2. The new particle diameter selected did not arrive at DMA2 while the current HV2 voltage sweep was running, but did arrive immediately after that sweep had been completed. DMA2 then immediately started the sweep on this new wet diameter. Thus the time taken to flush the tubing and the HUM is minimized. This reduced the time for performing HV2 sweeps on 12 different dry diameters to ~ 9 minutes.

Periodically, experiments were performed where a challenge aerosol of a pure inorganic salt (Sodium Chloride, NaCl, Ammonium Sulfate, $(NH_4)_2SO_4$, Potassium Iodide, KI or Potassim Chloride, KCl) was used to validate/update the calibration of DPH (as described in Suda et al., 2013). During an automated stepwise increase of HV2, the diameters $X_d$ and $X_w$ were precisely determined. The calculated saturation ratios (SR-calc) were obtained from knowledge of the dry diameter $X_d$, the wet diameter $X_w$ and the fact that the particles were a pure chemical of known properties. The SR-calc values were computed and compared to the value reported by the dew point hygrometer (SR-DPH). A calibration for the DPH was thus obtained. Typically, a value of 0.85 to 0.99 is obtained for SR-calc.

In the MST H-TDMA system, the SR is measured in the growth region by performing experiments (as recommended by Johnson et al., 2008). The SR is a function of not only the water vapor-air mixing ratio, but also a function of gas temperature. Even though the mixing ratio will not change as $Q_{s2}$ travels from the region of the DPH to the middle of DMA2, the temperature may, resulting in a potential change in SR. Thus, it is better to self-calibrate the H-TDMA system using this method. Furthermore, it is generally known that reliable measurements of SR from commercial instruments become very hard to obtain the closer one gets to SR=1.

All H-TDMA systems described in the literature are designed to provide precise values for the hygroscopic growth factor. Furthermore, almost all of these systems have the ability to vary the SR, thus requiring separate thermostating for the HUM and for DMA2 (Suda et al., 2013; Woods et al., 2013; Shi et al., 2012; Fors et al., 2010; Park et al., 2009a; Massling et al., 2011; Hu et al., 2010; Biskos et al., 2006; Lopez-Yglesias et al., 2014). Others (Johnson et al., 2008; Cubison et al., 2005) utilize controlled mixing of humid and dry air to achieve the desired humidity. Some systems include water baths (Hennig et al., 2005; Weingartner et al., 2002); temperature controlled cabinets (Cocker et al., 2001) and passive, insulated regions (Virkkula et al.,1999; Johnson et al., 2008).

Although these designs offer very good precision and the ability to vary the SR, they may not be well suited for field measurements, since most of them involve two separate volumes that must have their temperatures maintained very precisely. It is the temperature difference between these two volumes that is the critically important parameter. The MST H-TDMA system was designed to be less susceptible to ambient temperature fluctuations. This was achieved by encasing both the HUM and DMA2 in the same thermostated container (volume ~ 14 L). Other systems have also immersed DMA2 and the HUM in a water bath (Cubison et al., 2005; Hennig et al., 2005; Weingartner et al., 2002) to minimise the temperature gradients. In the MST H-TDMA system, temperature drifts are not critical, since the temperature difference between the HUM and the DMA2 (and the exposure time of the $Q_{p2}$ and $Q_{s2}$ in HUM) is what determines the SR, and that remains constant (zero temperature difference).

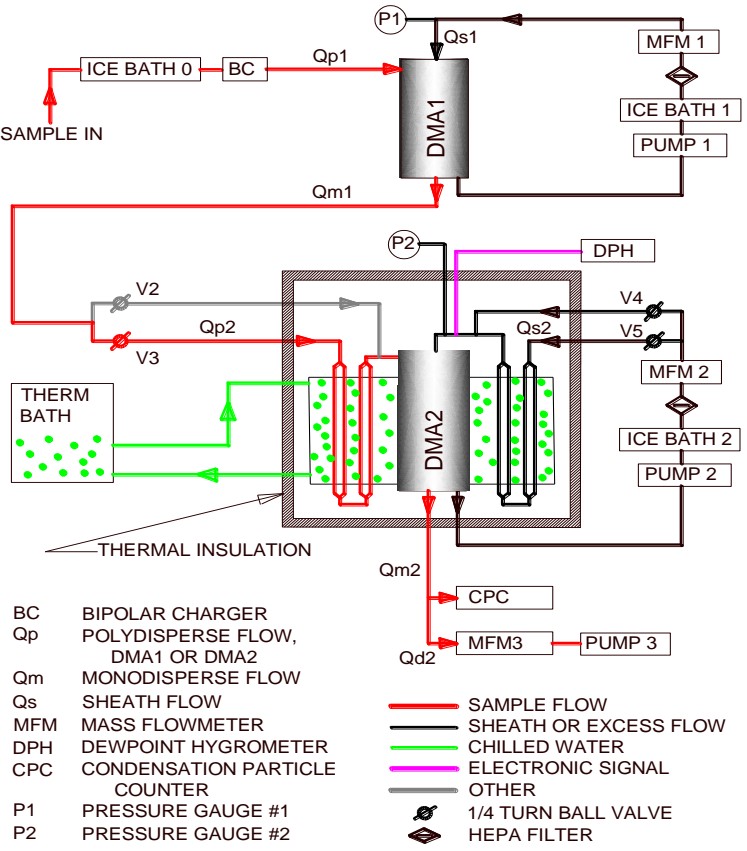

**Fig. 1.  Schematic of the MST H-TDMA system**

Suda et al., 2013 discussed the problem of DMA offset, whereby the diameter as measured by DMA1 may be slightly

5   different from the diameter as measured by DMA2, even if they both sample the same aerosol simultaneously.  This situation

was avoided in the MST H-TDMA system by performing a self-calibration.  To accomplish this, an inorganic challenge aerosol

(e.g. $(NH_4)_2SO_4$)) was delivered to DMA1 and LV directed DMA1 to deliver sample particles with a given diameter $X_d$ .  The

HUM was bypassed and LV initiated a voltage sweep on DMA2, which yielded a diameter $X_{wswp}$.  This was repeated for a

series of $X_d$ values ranging from 10 to 160 nm, establishing a calibration curve between $X_d$ and $X_{wswp}$ with $X_d$ taken as the true

10   diameter.  Within LV, this calibration was utilized to synchronize the two DMAs. Since DMA1 was static during a voltage

sweep and its $X_d$ involves no error from uncertainties in the lag time (LT2), DMA1 was chosen as the reference.

### 3.0 Hygroscopic Properties

#### 3.1 Determining the saturation ratio (SR)

The saturation ratio (SR) can be calculated from Köhler theory (Pruppacher and Klett, 1978). For hybrid particles that are composed of a spherical, insoluble core of diameter $X_u$ surrounded by a spherical shell of soluble material, SR can be calculated from:

$$\ln(SR) = \frac{2A}{X_w} - \frac{8B}{(X_w^3 - X_u^3)} \tag{1}$$

where $X_w$ is the diameter of the solution droplet. By expanding ln (SR) in a Taylor series and keeping only the first term in the expansion, an error of less than 4.5% is introduced. Thus Eq. (1) can be approximated as

$$SR = 1 + \frac{2A}{X_w} - \frac{8B}{(X_w^3 - X_u^3)} \tag{2}$$

$$A = \frac{(2\,M_w\,\sigma_{w/a})}{R\,T\,\rho_w} \sim \frac{(3.12*10^2\,nm*K)}{T} = \frac{3.12 \cdot 10^{-7}\,m \cdot K}{T} \tag{3}$$

$$B = \left(\frac{4.297 \cdot 10^{-6}\,m^3}{mol}\right)\frac{\nu\,m_s\,\phi_s}{M_s} \tag{4}$$

where $M_w$ is the molecular weight of water, $\sigma_{w/a}$ is the surface tension of the solution/air interface ($7.2 \cdot 10^{-2}$ N m$^{-1}$), R is the universal gas constant [8.31 (N m K$^{-1}$ mol$^{-1}$)], T is the absolute temperature, $\rho_w$ is the density of water, $\nu$ is the number of ions into which the solute material disassociates, $m_s$ is the mass of the dry (salt or solute) particle, $\Phi_s$ is the osmotic coefficient of the solution droplet, and $M_s$ is the molecular weight of the solute.

For particles composed of a single, pure chemical species with no insoluble core ($X_u=0$)

$$SR = 1 + \frac{2A}{X_w} - \frac{8B}{(X_w^3)} \tag{5}$$

and A and B remain as defined above. The mass of the dry (salt or solute) particle is given by:

$$m_s = \left(\frac{\pi}{6}\right)\rho_s\,(X_d^3) \tag{6}$$

The osmotic coefficients for selected solute materials as a function of the molality has been reported in the literature (Hamer et al., 1972; Robinson et al., 2002; Staples, 1981). $\Phi_s$ can be related to the square root of the molality ($\psi$) by a 6$^{th}$ order

polynomial function. Hence $\Phi_s$ is dry and wet diameter dependent, and this must be taken into account. The molality ($\psi$) (number of moles of the solute / mass of solvent in kg) is given by:

$$\psi = \frac{n(solute)}{mass\ of\ solvent(kg)} = \frac{1000\left(\frac{\pi}{6}\right)\rho_S(X_d^3)M_S^{-1}}{\left(\frac{\pi}{6}\right)\rho_w(X_w^3 - X_d^3)} = \frac{1000\rho_S(X_d^3)}{\rho_w M_S(X_w^3 - X_d^3)} \tag{7}$$

where n is the number of moles of the solute. Examples of how $\Phi_s$ is determined are provided in the supplemental information Thus a pure chemical of known properties can be used to self-calibrate the H-TDMA and verify SR.

### 3.2 Determining the water activity factor, $a_w$

The Köhler theory (Pruppacher and Klett, 1978) describes how the saturation ratio (SR) over an aqueous solution droplet is related to other parameters characterizing the water droplet.

$$SR = a_w \cdot \exp\left(\frac{4*(\sigma_{w/a/})*M_w}{R*T*\rho_w*X_w}\right) \tag{8}$$

where $a_w$ is the activity of water in solution, and $X_w$ is the diameter of the droplet determined by the voltage sweep of DMA2/CPC. Thus $a_w$ can be calculated from Eq. (8).

### 3.3 Determining the Growth Factor

The growth factor (GF) is the most commonly used parameter to describe the hygroscopic properties of particles. It is defined

as:

$$GF = \frac{X_w}{X_d} \tag{9}$$

where $X_w$ is the wet particle diameter and $X_d$ is the dry particle diameter. GF is a function of SR and provides a measure of the relative change in size of the particle as a result of water absorption.

**3.4 Determining the hygroscopicity parameter ($\kappa$)**

    Petters and Kreidenweis (2007) proposed that a single parameter representation for hygroscopicity was better to model complex, multicomponent particles types such as atmospheric particles containing insoluble components. The hygroscopicity parameter ($\kappa$) is defined through its effect on the water activity of the solution by:

$$\frac{1}{a_w} = 1 + \kappa\left(\frac{V_{solute}}{V_{water}}\right) \tag{10}$$

where $V_{solute}$ is the volume of the dry particulate matter and $V_{water}$ is the volume of the water. It should be noted that $V_{solute}$ also includes the volume of the insoluble core, if there is one. For clarity, we note that

$$V_{solute} = \frac{\pi}{6} (X_d^3) \tag{11}$$

$$V_{water} = \frac{\pi}{6} (X_w^3 - X_d^3) \tag{12}$$

The $\kappa$, calculated from Eq (10), is an excellent choice when studying ambient aerosols that derive from the agglomeration of particles from multiple sources. It should be noted that $\kappa$ can also be calculated from the GF and $a_w$ without determining the wet and dry volumes (Holmgren et al., 2014).

$$\kappa = (GF^3 - 1) \times (1 - a_w)/a_w \tag{13}$$

Thus, for an aerosol of unknown composition, Eq. (8) is used to compute $a_w$, Eq. (9) for GF, and then Eq. (13) to compute $\kappa$. It should also be noted that for an aerosol of unknown composition, only equations 8-13 are used, and none of these require any prior knowledge of the physical or chemical properties of the aerosol.

## 4 MST H-TDMA performance evaluation

### 4.1 Performance evaluation using pure inorganic salts

The performance of the MST H-TDMA system was evaluated by measuring GF of pure inorganic salts and comparing them to theory. The values of GF vs. $X_d$ were measured and plotted for NaCl, $(NH_4)_2SO_4$, KI and KCl in Fig. 2. To obtain the theoretical GF, the SR-calc (Eqs. 3-6) for the largest two or three dry particle diameters was computed and an average was obtained. This SR-calc value was then used to compute the theoretical GF for the smaller particle diameters. There is excellent agreement between the measured growth factor and the value predicted from theory. It should also be noted that the osmotic coefficient $\Phi_s$ is quite different from unity in several of the cases.

The dry diameter estimate ($X_d$) requires a knowledge of the average particle diameter actually exiting DMA1. A weighted average (neglecting doubly charged particles) is given by:

$$X_d = X_{avg} = \sum_{k=1}^{N} ( SNN_k * X_k * F_k * TF_k * dlogX_k ) / ( SNN_k * F_k * TF_k * dlogX_k ) \tag{14}$$

where $SNN_k$ is the differential size distribution entering the H-TDMA system (measured here by a Cambustion DMS500), $X_k$ is the particle diameter, $F_k$ is the fraction of particles of diameter $X_k$ that carry one elementary charge (Hagen et al., 1983), $TF_k$ is the transfer function of DMA1, and $dlogX_k$ is the differential in $logX$ between adjacent data points in $SNN_k$.

The use of Eq. (14) rather than the DMA1 set point value for the average particle diameter provided a more accurate $X_d$ value for these pure chemicals. The DMS500 reported the peak in $SNN_k$ was at approximately 27 nm for the nebulizer and the solutions of pure solute chemicals used. Since $SNN_k$ and $F_k$ were both monotonically increasing over the range where $TF_k$ was non-zero, the $X_{avg}$ was greater than what DMA1 was tuned to. For example, when DMA1 was set to extract particles with $X_d$ = 12.76 nm, the value of $X_{avg}$ from Eq. (14) was found to be 13.49 nm which resulted in a change to the GF from 2.33 to 2.22 (a 5% correction). This correction was taken into account for particle diameters less than 20 nm. For particles diameters larger than 20nm, the correction is insignificant. This correction can be utilized for any diameter $X_d$ as long as the $SNN_k$ , the $F_k$, and the $TF_k$ are known.

Most H-TDMA systems for which data is reported in the literature are designed to scan the SR (called humidigrams) and report (1) the GF for a wide SR range (0.20 < SR < 1), and (2) the deliquescence relative humidity, i.e., the SR at which the dry particles very abruptly begin to take on liquid water and grow to a much larger solution droplets. The MST H-TDMA system was not designed to perform humidigrams. By inspection of humidigrams in the literature and with knowledge of the SR that was recorded in the MST H-TDMA, the GF from these other systems can be estimated. Figs 2 (a)-(d) present the experimentally obtained GF as a function of $X_d$ for various inorganic salts. The theoretical values along with those reported in the literature from other systems are in good agreement with the GF determined by the MST H-TDMA. .

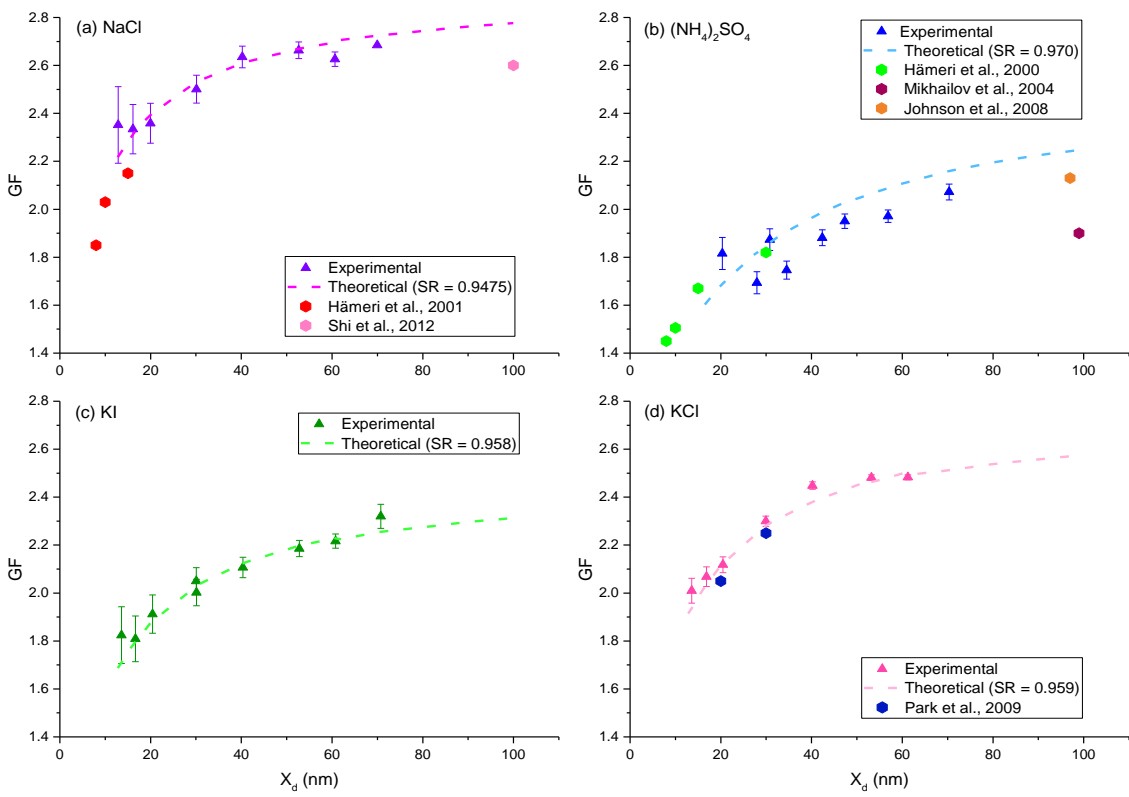

**Fig. 2. Growth factor as a function of dry particle diameter ($X_d$) for NaCl (a), $(NH_4)_2SO_4$ (b), KI (c), and KCl (d).**

5   Fig. (3) is a plot of κ vs. $X_d$ for the same four chemicals. Also plotted are the ranges of κ values for $(NH_4)_2SO_4$ and NaCl as reported by Petters and Kreidenweis (2007), Clegg et al., (1998), and Koehler et al., (2006). There is good agreement between the κ values reported by the MST H-TDMA system and those from literature.

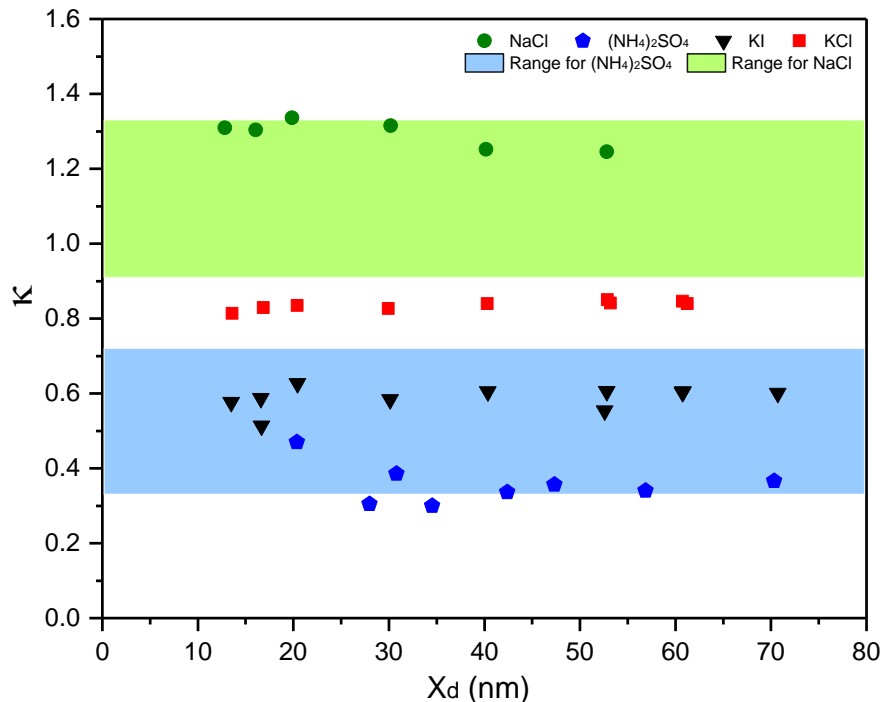

**Fig. 3. Hygroscopicity parameter ($\kappa$) as a function of dry particle diameter ($X_d$) for NaCl (a), $(NH_4)_2SO_4$ (b), KI (c), and KCl (d).**

### 4.2 Residence time

Since the deliquescence technique is an equilibrium based methodology, the closeness to equilibrium must be validated, especially for the larger droplets (which grow more slowly). For such a test, the H-TDMA system was configured to select a dry diameter ($X_d$ = 17 nm, 30 nm, or 51 nm) of $(NH_4)_2SO_4$ aerosol. The wet diameter ($X_w$) was measured, allowing calculation of GF and SR-calc. This was repeated for a series of $Q_{p2}$ values, which varied the residence time. The results are shown in Fig. 4.

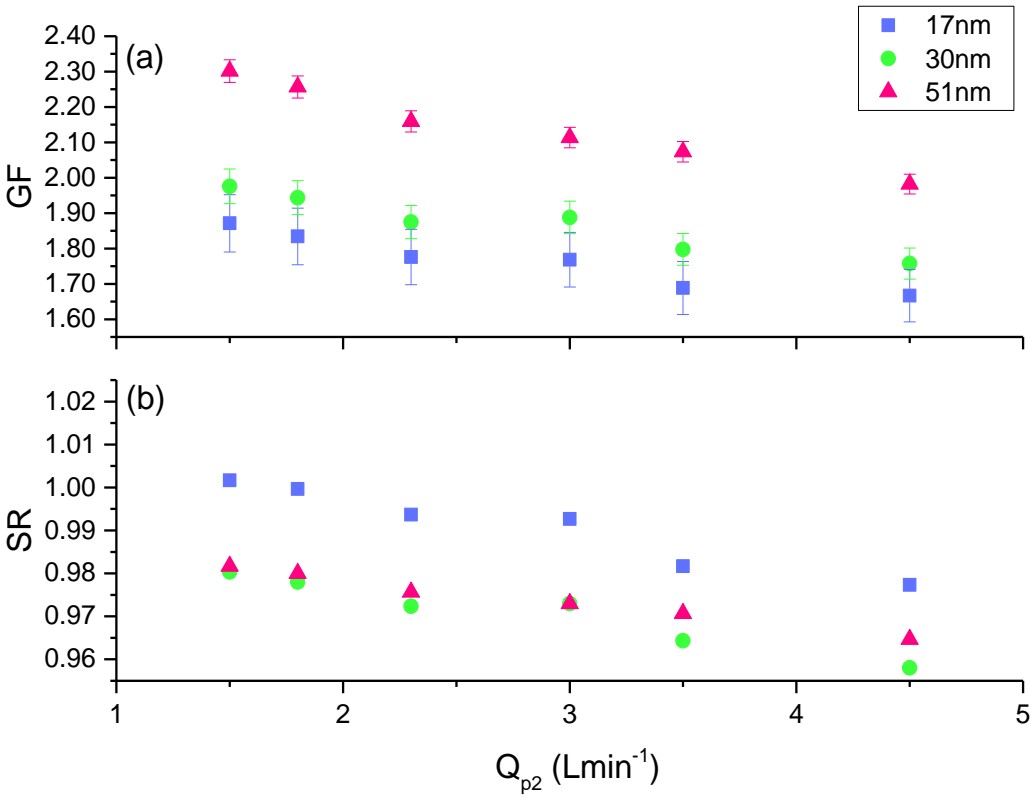

**Fig. 4.** **Growth Factor (GF) (a), and Saturation Ratio (SR-calc) (b) as a function of polydisperse flowrate $Q_{p2}$, with challenge $(NH_4)_2SO_4$ aerosols of 17 nm, 30 nm, and 51 nm.**

From Fig. 4. (a) and (b), a small dependence of GF and SR-calc on $Q_{p2}$ is observed. Utilizing a small $Q_{p2}$ would be best to achieve the highest SR. However, very small values of $Q_{p2}$ result in very low concentration delivered to the CPC. In field measurements where the sample is diluted with ambient air, the concentration is already quite low leading to signal to noise issues. Alternatively, at large values of $Q_{p2}$, the peak is too broad. To avoid both of these extremes the H-TDMA system was operated at $Q_{p2} = 3.0$ L min$^{-1}$.

The H-TDMA system, when deployed in the field, is primarily intended to study particles with small $X_d$ values and small GFs. These particles will probably not grow large enough to experience insufficient growth time problems. However, it is good practice to periodically check the system and the sample aerosol by choosing a large $X_d$ (30 nm or larger) to determine if changes to $Q_{p2}$ result in a change to SR. If this is the case, then it is better to maintain $Q_{p2}$ at a lower value (2.0 L min$^{-1}$).

**4.3 Stability over long operating times**

For field applications, the H-TDMA system is required to maintain stable operation for long periods of time. The HUM tubes are wetted at the beginning of the day and need to be periodically rewetted to maintain a stable SR. The time after which the HUM tubes need to be rewetted was experimentally determined. Fig. 5 displays the results of determining the SR-calc by using particles of pure $(NH_4)_2SO_4$ ($X_u=0$ in Eq. (1)) and measuring the wet diameter $X_w$, given that the dry diameter set in DMA1 is held constant. Experiments were performed where the HUM tubes were wet thoroughly, and then automated scans were conducted for several hours with no further tube wetting. After the experimental measurements were performed, the SR was calculated from Eq. (5). Also shown is the measured SR of the $Q_{sh2}$ as determined by DPH. Fig. 5 shows that the calculated and the measured SR remained constant for a period of over 225 minutes without having to rewet the tubes.

When required, tube rewetting was accomplished using a LabVIEW program which acted through a relay board to energize a peristaltic pump and sequentially opened twelve pinch valves for a short period (set by the operator), allowing each tube to be rewet in sequence. After rewetting, valves at the bottom of the 12 stainless steel tubes were manually opened to allow excess water to drain. During normal operations in the field, the HUM tubes were rewetted every 150 minutes.

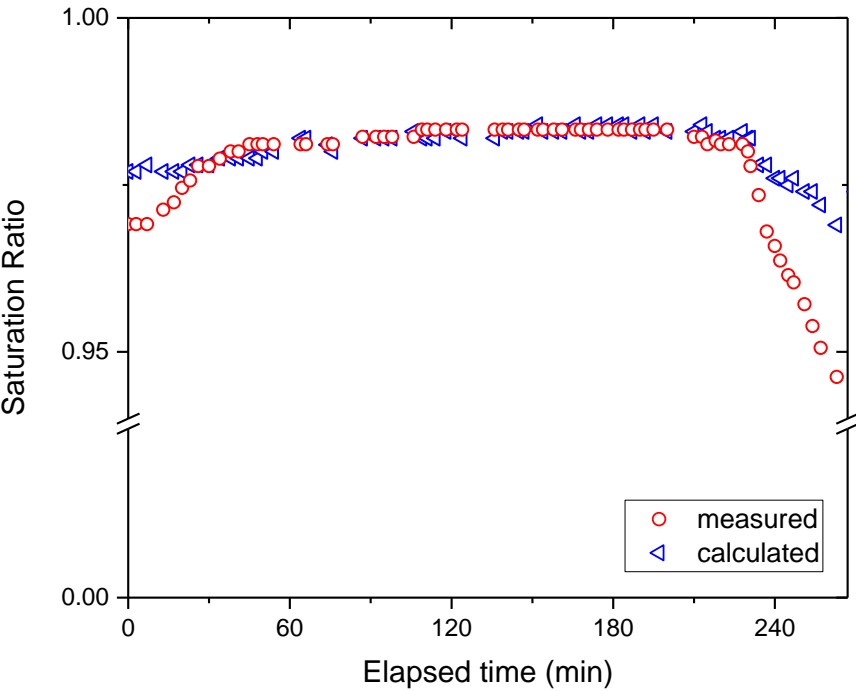

**Fig. 5. SR as a function of elapsed time since last wetting for pure particles of $(NH_4)_2SO_4$. The uncertainty in the SR (calculated) is approximately 0.008.**

### 4.4 Stability over varying ambient temperature conditions

The H-TDMA must be able to operate under varying ambient temperature conditions in the field.  The stability of the H-TDMA system was assessed using pure $(NH_4)_2SO_4$ as the challenge aerosol. DMA1 was set to extract dry particles of 30 nm. An automated voltage sweep with DMA2 was performed every 2 minutes, to determine $X_w$. The SR-calc was computed using Eq. (1), with $X_u = 0$.  At t=20 min (and 40 min), the ambient conditions surrounding the H-TDMA system were abruptly changed by blowing cold air over the bottom of the HUM tubes (or not blowing cold air over the bottom of the HUM), which is not as well thermally insulated as the rest of the H-TDMA system (Fig. 1).  This experiment was repeated four times on four different days.  The SR-calc remained constant over the duration of any one run as shown in Fig. 6. The average standard deviation in SR-calc divided by the average SR-calc for that trial over all four trials (120 measurements) was 0.0019, indicating that this system was insensitive to ambient temperature fluctuations.

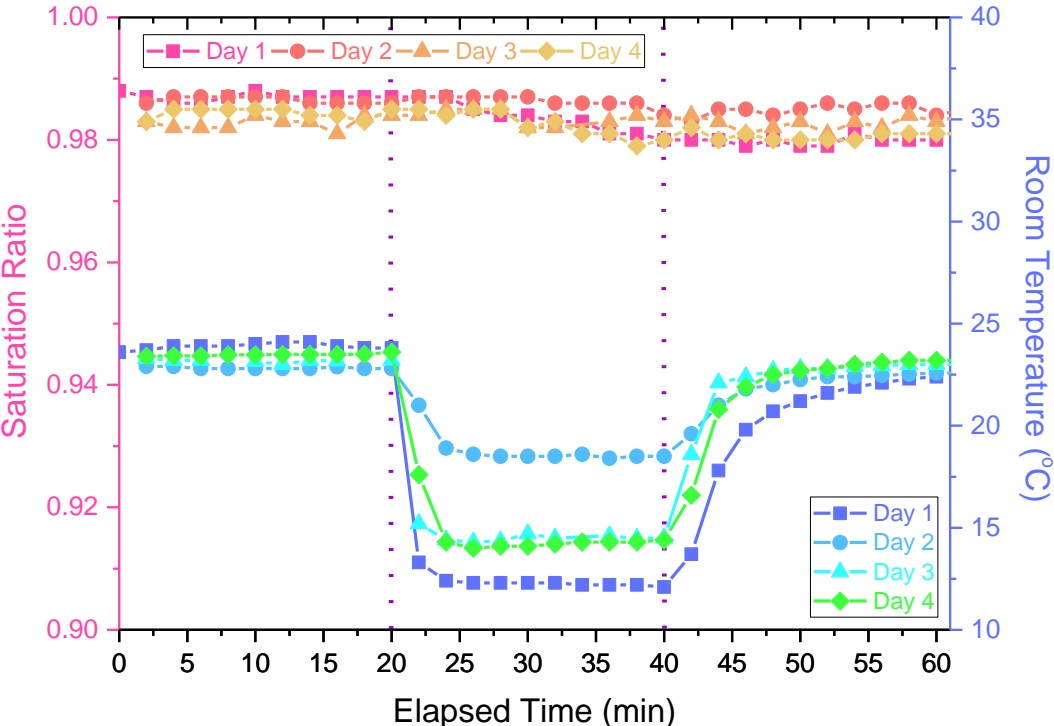

**Fig. 6.  Saturation Ratio (SR-calc) and Room Temperature as a function of elapsed time.**

# 5 Field deployment during the AAFEX II campaign

The MST H-TDMA system was deployed as part of the Alternative Aviation Fuels EXperiment (AAFEX II) campaign conducted during 20 March - 2 April 2011 at the NASA Dryden Aircraft Operations Facility (DAOF), Palmdale, CA, USA. The NASA DC-8 aircraft equipped with CFM56-2C1 engines was utilized as the emissions source. The aircraft was parked in an open air run-up facility with no other aircraft or emission sources in the vicinity of the test site. Detailed descriptions of the test site and experimental set up have been previously reported (Timko et al., 2013; Moore et al., 2015). The main objective of the campaign was to investigate the gaseous and PM emissions characteristics of the CFM56-2C1 engine burning conventional and alternative fuels as a function of engine thrust conditions at several sampling locations in the exhaust plume. PM emissions data were acquired for a typical cycle which consisted of the following engine thrust conditions: 4%, 7%, 30%, 65%, 85% and 100% rated thrust. Two test cycles were run for each fuel – one stepping up from 4% to 100% rated thrust, and the other stepping down from 100% to 4% rated thrust. Five fuels were used during the campaign: (1) JP-8 (the military equivalent of conventional Jet A/JetA-1), (2) tallow-based hydro-processed esters and fatty acids (HEFA), (3) coal-derived Sasol Fischer-Tropsch (FT), (4) a blend of HEFA and JP-8, and (5) FT doped with Tetrahydrothiophene (THT) to boost the sulfur content of the fuel. A summary of selected fuel properties is provided in Table 1. Chemical and physical analysis of the HEFA and FT fuels has been reported elsewhere (Corporan et al., 2011).

**Table 1.** Selected fuel properties

| Property | Method | JP-8 | HEFA | FT | HEFA-JP-8 Blend | FT+THT |
|---|---|---|---|---|---|---|
| Density @ 15°C (kg$^{-1}$) | ASTM D4052 | 0.811 | 0.758 | 0.761 | 0.783 | 0.761 |
| Viscosity @ -20°C (mm²s$^{-1}$) | ASTM D445 | 4.1 | 4.9 | 3.7 | 4.3 | 3.2 |
| Distillation temperature (°C) | ASTM D86 | | | | | |
|   10% recovered | | 168 | 175 | 164 | 166 | 164 |
|   End point | | 268 | 254 | 226 | 263 | 224 |
| Flash Point (°C) | ASTM D93 | 46 | 52 | 43 | 46 | 43 |
| Net Heat of Combustion (MJkg$^{-1}$) | ASTM D4809 | 42.8 | 43.6 | 43.8 | 43.3 | 43.8 |
| Aromatics (% vol) | ASTM D1319 | 21.8 | 0.4 | 1.4 | 10.2 | 2.1 |
| Naphthalenes (% vol ) | ASTM D1840 | 1.3 | 0 | 0 | 0.65 | 0 |
| Sulphur (ppm) | ASTM D2622 | 188 | 6 | 4 | 276 | 1083 |
| Hydrogen Content (% mass) | ASTM D3343 | 13.5 | 15.3 | 15 | 14.4 | 15 |

| Carbon content (% mass) | calculated | 86.5 | 84.7 | 85 | 85.6 | 85 |
| H/C ratio | calculated | 1.86 | 2.15 | 2.10 | 2.00 | 2.10 |

The emissions from the CFM56-2C1 engine were measured at several distances (1m, 30m, and 143m) from the engine exit plane to study the PM characteristics as the exhaust plume cooled and mixed with ambient air. Only data acquired at the 143 m location are presented and discussed here to investigate the hygroscopic properties of the evolving plume.

A 2 inch ID aluminum tube (~ 1.3 m above the concrete apron) positioned downwind from #3 engine on the starboard side of the aircraft was used to extract exhaust plume samples at the 143m location. The exhaust was transported to a small trailer approximately 18 m away which housed the MST H-TDMA system to measure hygroscopic properties. The exhaust gas flow rate through the 2 inch ID x 18 m L tubing was well over 100 L min$^{-1}$. Also housed in the trailer was a Cambustion DMS500 (Reavell et al., 2002; Hagen et al., 2009) which measured the real-time particle size distributions, and a LI-COR 840A NDIR detector that measured exhaust $CO_2$ concentration. Ambient meteorological conditions such as temperature, pressure, and relative humidity were also monitored and recorded throughout the campaign. The exhaust samples at 4% and 7% engine thrust conditions were impacted by the ambient conditions, specifically, wind speed and wind direction. However, the $CO_2$ measurements during the 7% trust periods were approximately twice the background level, indicating that the exhaust plume was being sampled.

The DMS500 measured total PM size distributions. The nvPM size distributions were obtained by passing the sample through a thermal denuder. The thermal denuder consisted of a coil of stainless steel tubing (0.457cm ID) housed in a temperature controlled aluminium box heated to 300°C, followed by a cooling section. It is similar in design to that used by Saleh et al. (2011), and has been used in a previous study (Rye et al., 2012). Laboratory evaluations have demonstrated that $H_2SO_4$ droplets of diameter 10 – 100 nm are almost completely evaporated in the thermal denuder.

The total and nvPM number-based size distributions were converted to number-based emission index (EIn) distributions to account for varying amounts of dilution for each plume, and are presented for selected fuels at the 100% thrust condition shown in Fig. 7. The total PM size distributions are bi-modal with a strong nucleation mode (<20 nm) and an accumulation mode. These observations are consistent with those reported for PM emissions measured downwind of several different aircraft engine types (Lobo et al., 2007; Lobo et al., 2012; Lobo et al., 2015a). The enhancement of the nucleation mode in measurements made downwind of the engine exit plane is due to gas-to-particle conversion in the exhaust plume driven by fuel composition, ambient conditions, and degree of mixing. Timko et al., 2013, found that the driving force for gas to particle conversion in the expanding exhaust plume was the ratio of particle precursors (both organic and sulfate) to soot.

The sulfur in the fuel is oxidized to $SO_2$, which then undergoes oxidation to $SO_3$ and subsequently to sulphuric acid ($H_2SO_4$) in the exhaust plume (Miake-Lye et al., 1998; Schumann et al., 2002). The $H_2SO_4$ either homogenously nucleates to form pure $H_2SO_4$ droplets, or condenses onto existing soot particles to form hybrid particles that have significant water soluble components (Gysel, et al., 2003; Wyslouzil et al., 1994).

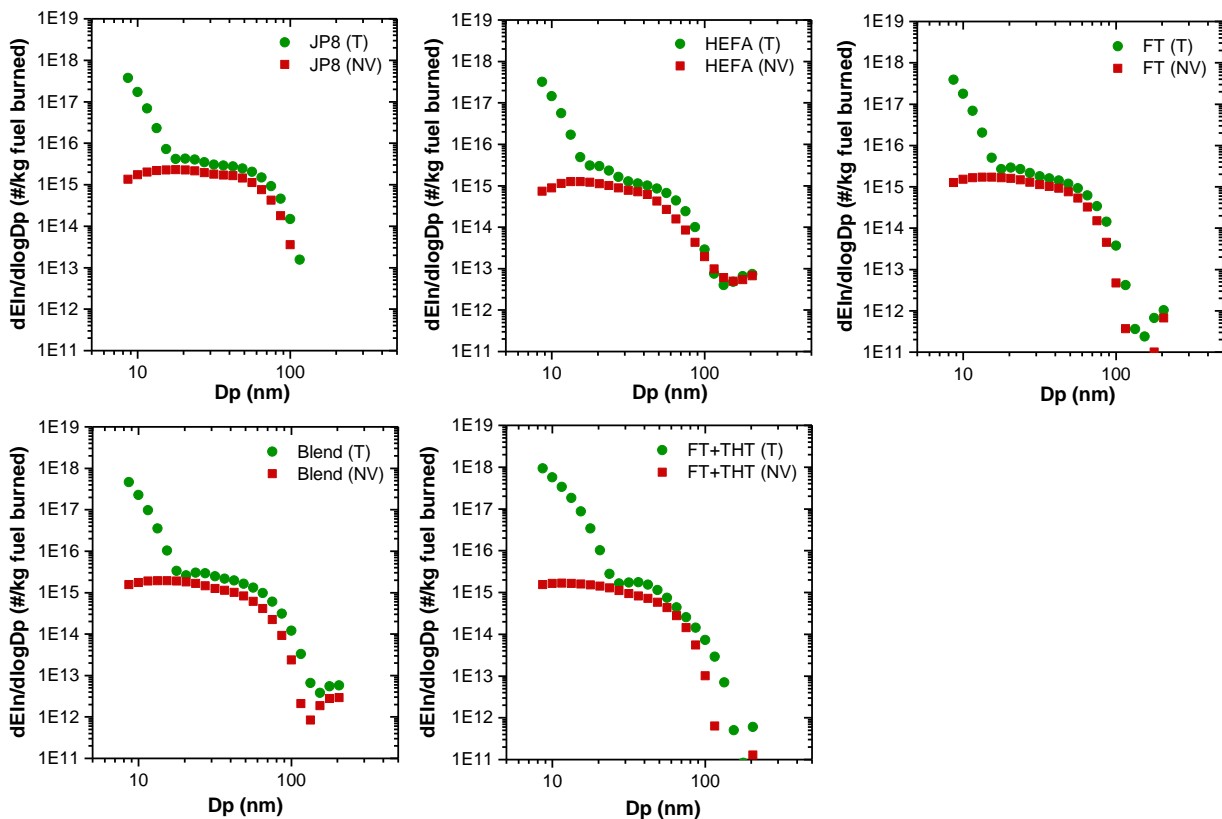

**Fig. 7. Total (T) and non-volatile (NV) PM number-based emission index (EIn) size distributions for the various fuels at the 100% engine thrust condition.**

The data acquired with the MST H-TDMA system was used to calculate GF and κ of these particles as a function of fuel type, engine thrust condition, and dry particle diameter. The H-TDMA was operated with a SR of 0.91. Fig. 8 shows GF and κ as a function of $X_d$ for particles generated at different engine thrust conditions and different fuels. The uncertainty in GF was 9% for particles with diameter ~10 nm, and 3% for the larger diameters (26 nm). The uncertainty in κ was 7% and 2% for particles with diameter ~ 10 nm and ~26 nm, respectively.

Gysel et al. 2007, state that $H_2SO_4$ is expected to retain water at 5-10% RH, corresponding to a growth factor of ~ 1.15, and took this into account when calculating the mixed particle growth factor in their data. This procedure was similarly followed for the current dataset. Thus the measured $X_d$'s were scaled by a factor of 0.869.

    For a given engine thrust condition, both GF and κ increased with increasing fuel sulfur content. GF and κ were also observed to be dependent on particle diameter, withthe highest GF and κ for particles ~10 nm, and decreasing for large particle diameters.

This increase in GF and κ corresponds to the nucleation mode in the size distributions (Fig 7), which was composed of particles or droplets formed by the homogeneous nucleation of low equilibrium vapor pressure species, such as $H_2SO_4$ and other water

soluble organic compounds. The GF and κ were also found to increase with increasing engine thrust condition for a given $X_d$, with the largest values observed at the 100% engine thrust condition.

Gysel et al. 2003, reported GF of particles from a jet engine combustor burning three different fuels with 50 ppm, 410 ppm, and 1270 ppm of sulfur at two inlet temperature operating conditions: 566 K and 766K. These data are in good agreement with the current study for very low sulfur (HEFA and FT) fuels, conventional JP-8, and the sulfur enhanced FT (FT+THT), respectively.

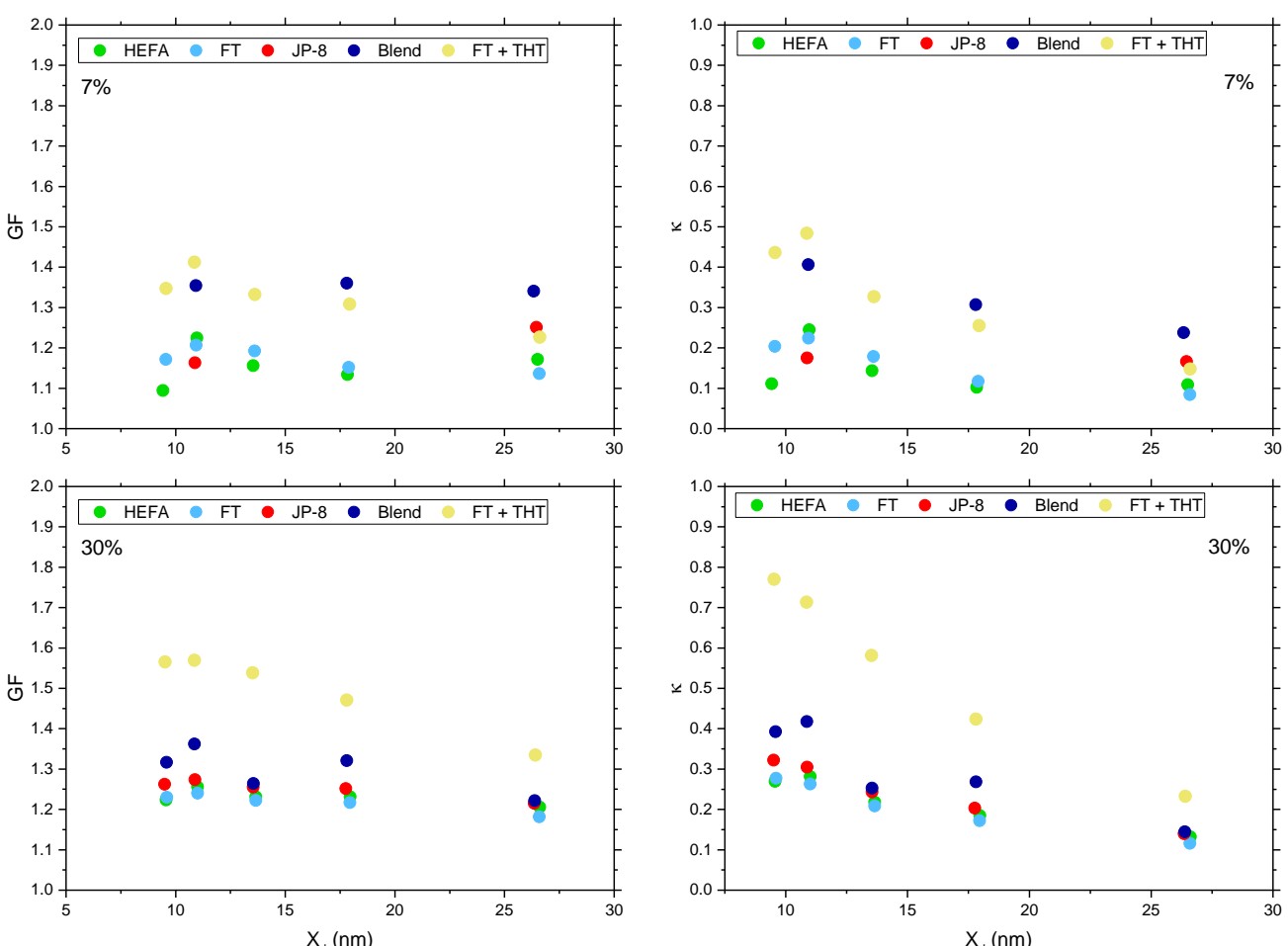

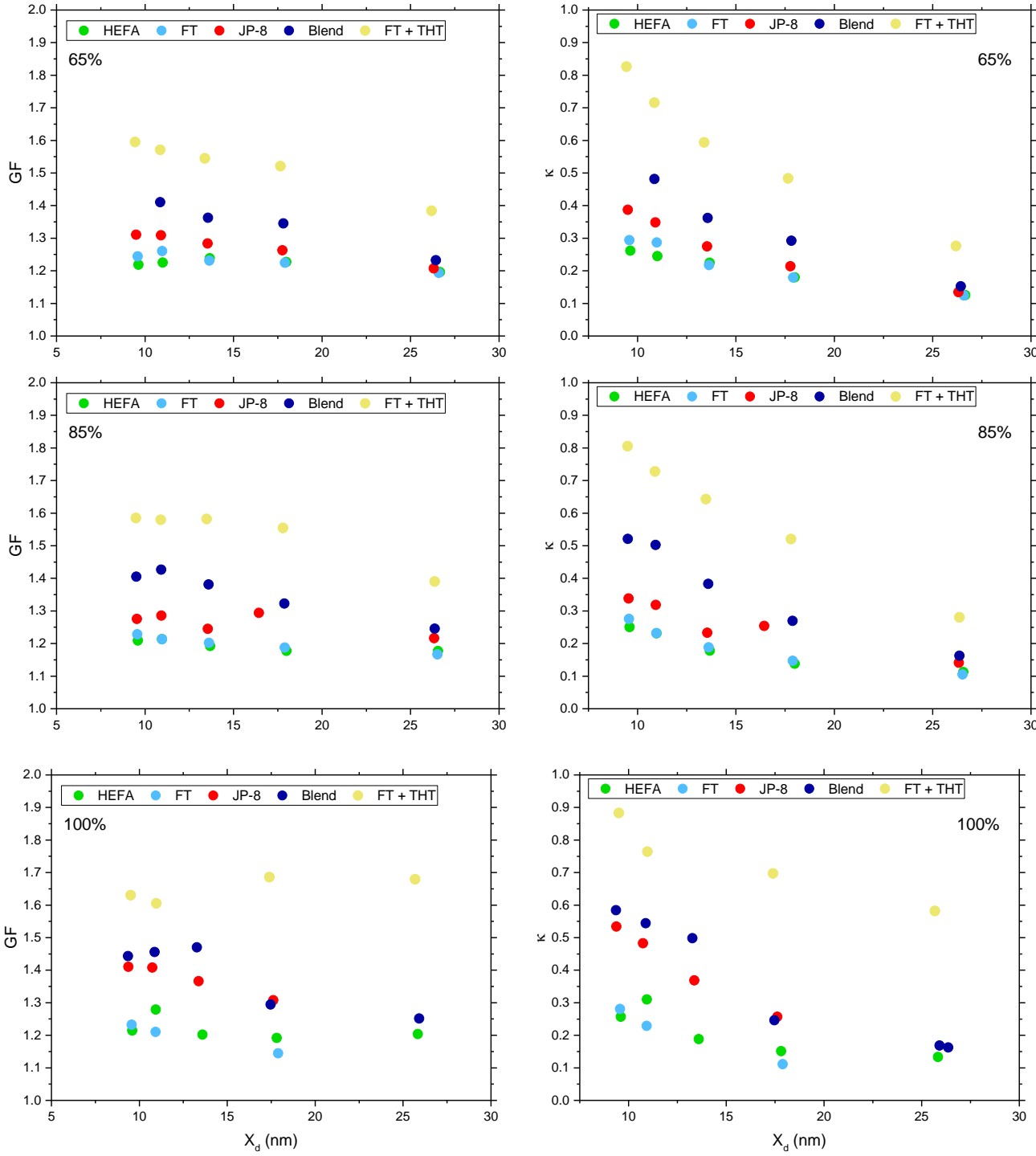

**Fig. 8. GF and κ as a function of $X_d$ for particles generated at different engine thrust conditions and different fuels.**

## 6 Conclusions

A robust, mobile H-TDMA system has been developed for field measurements that involve (1) sources that are very expensive to operate, (2) exhaust plumes influenced by wind speed and direction, and (3) varying meteorological conditions. The GF exhibited by particles of four inorganic salts was studied and found to be in good agreement with theory and with other experimental data reported in the literature. The fixed SR provided by the H-TDMA system during laboratory evaluation (typically ~ 0.98) was found to be quite constant over long periods of time, even when the ambient temperature varied considerably, making the MST H-TDMA system suitable for field experiments. The H-TDMA was demonstrated to perform a scan to determine GF and κ for one dry diameter in approximately 45 s. It performed scans over as many as 12 dry diameters sequentially in ~ 9 min. The H-TDMA system provided parameterization for hygroscopic properties for aircraft engine exhaust plumes in terms of GF and κ during the AAFEX II field campaign. It was observed that GF and κ: (1) increased with fuel sulfur content, (2) increased with increasing engine thrust condition, and (3) decreased with increasing dry particle diameter.

*Acknowledgements*

This work was partly funded by the US Federal Aviation Administration (FAA) through the Partnership for AiR Transportation for Noise and Emissions Reduction (PARTNER) – an FAA-NASA-Transport Canada-US DoD-US EPA sponsored Center of Excellence Project 20 under Grant No. 09-C-NE-MST Amendment 003. Any opinions, findings, and conclusions or recommendations expressed in this paper are those of the authors and do not necessarily reflect the views of the FAA. We thank the entire AAFEX II project team for their support during the campaign. Dr Otmar Schmid performed many early experiments to validate the worthiness of this device and provided impetus for continued effort to develop this instrument. We thank Veronica Villines Teat, Emitt Witt, Christian Hurst, Nicholas Altese, Elizabeth Black, and Jonathon Sidwell for their assistance in gathering some of the data. We are also grateful to Dr. Markus Petters and Dr. Sonia Kreidenweis for their assistance with the κ calculations.

**List of Abbreviations**

| | |
|---|---|
| AAFEX | Alternative Aviation Fuels EXperiment |
| ASTM | American Society for Testing and Materials |
| BC | Bipolar Charger |
| CPC | Condensation Particle Counter |
| DPH | Dew Point Hygrometer |
| DMA | Differential Mobility Analyzer |
| DRH | deliquescence relative humidity – the humidity at which the dry particles abruptly take on water and become solution drops |

| | FT | Fischer-Tropsch |
|---|---|---|
| | GF | growth factor, $X_w/X_d$ |
| | H-TDMA | Hygroscopicity Tandem Differential Mobility Analyzer |
| | HEFA | hydro-processed esters and fatty acids |
| 5 | HUM | humidifier |
| | HV1, HV2 | high voltage in DMA1 or DMA2 |
| | IB | Ice Bath |
| | LV | LabVIEW program |
| | MST | Missouri University of Science and Technology |
| 10 | nvPM | non-volatile particulate matter |
| | PM | particulate matter |
| | R | Ideal Gas Law constant |
| | SR | Saturation Ratio |
| | SR-calc | value of SR calculated from measured values of $X_d$, $X_w$, when using a pure salt |
| 15 | SR-DPH | value of SR measured by the Dew Point Hygrometer |
| | THT | tetrahydrothiophene |

**List of Symbols**

| | Symbol | Units | Quantity |
|---|---|---|---|
| 20 | dt | s | elapsed time since a trial run began |
| | $dt_{max}$ | s | value of dt when CPC reading is at its maximum |
| | $F_k$ | | fraction of particles of diameter $X_k$ that carry one elementary charge |
| | LT2 | s | lag time between when a voltage is imposed on DMA2 and when the particles selected by that voltage reach the CPC |
| 25 | $M_s$ | g mole$^{-1}$ | molecular weight of solute |
| | $M_w$ | g mole$^{-1}$ | molecular weight of water |
| | $m_s$ | g | mass of water soluble portion of the dry particle |
| | P1, P2 | psia | pressure in $Q_{s1}$, $Q_{s2}$ flow in either DMA1 or DMA2 |
| | $Q_{p1}$, $Q_{p2}$ | L min$^{-1}$ | polydisperse aerosol gas flow rate, either for DMA1 or DMA2 |
| 30 | $Q_{s1}$, $Q_{s2}$ | L min$^{-1}$ | sheath gas flow rate, either for DMA1 or DMA2 |
| | $Q_{m1}$, $Q_{m2}$ | L min$^{-1}$ | monodisperse aerosol gas flow rate, either for DMA1 or DMA2 |
| | $Q_d$ | L min$^{-1}$ | flow rate of dump gas in parallel with the CPC |
| | $SNN_k$ | | differential size distribution entering the H-TDMA system |
| | T | K | absolute temperature |

| | | |
|---|---|---|
| $TF_k$ | | value of transfer function of DMA1 for k-th point in the series to determine $X_{avg}$ |
| $X_{avg}$ | nm | average particle diameter exiting the dry DMA, DMA1 |
| $X_d$ | nm | set point diameter of DMA1 |
| $X_u$ | nm | diameter of insoluble core in hybrid particle |
| $X_w$ | nm | diameter of wet particle or solution droplet formed from dry particle after passing through the HUM |
| $X_{wswp}$ | nm | diameter of particles (solution drops) exiting DMA2 as measured by LV doing an automated sweep |
| $X_k$ | nm | the k-th particle diameter in the series to determine the $X_{avg}$ |
| $\Psi$ | moles kg$^{-1}$ | molality of the solution droplet |
| $\nu$ | | number of ions into which the soluble salt disassociates |
| $\Phi_s$ | | Osmotic coefficient of the solution droplet |
| $\rho_s$ | g cm$^{-3}$ | density of soluble material in hybrid particle |
| $\rho_w$ | g cm$^{-3}$ | density of water |
| $\sigma_{w/a}$ | N m$^{-1}$ | surface tension of water against air |

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
