# Peer review of "Application of a Hygroscopicity Tandem Differential Mobility Analyzer for characterizing PM Emissions in exhaust plumes from an Aircraft Engine burning Conventional and Alternative fuels"

_Atmospheric Chemistry and Physics, 2018_

## Referee Comment (RC1) · Anonymous Referee #1 · 25 Jul 2018

This manuscript describes the design and performance evaluation of a Hydroscopic-Tandem Differential Mobility Analyzer (H-TDMA) as well as the subsequent field deployment to measure soluble mass fraction of aircraft engine PM emissions from CFM56-2C1 engines burning several fuels during the AAFEX II campaign. As the authors specify, this H-TDMA was designed for (1) field measurements involving mobile sources that are very costly to operate, (2) when exhaust sample plumes are available for only short periods of time, and (3) for varying ambient conditions.

Overall, the authors do a good job of describing in details the operation and characterization of the instrument. This study represents a substantial contribution to the field of aerosol hydroscopic property measurements. I recommend it be accepted for publication in Aerosol Science and Technology after minor revisions.

In addition, I include the following comments to help improve the readability and clarity of the manuscript:

1. There are a large number of abbreviations in this manuscript. I would recommend to include a list of abbreviations at the end of the text.

2. Page 3, line 12: In the text, the authors use the unit "L m-1" as the abbreviation of liter per minute. Since "m" is also used as the abbreviation for meter in many places in this manuscript, I would recommend to change the unit of flow rate to "L min-1".

3. Page 7, line 9: The authors claim that u and s are assumed to be known, but they only provide the assumed value of u based on a previously published study. What is the assumed value of s used in this study?

4. Equation (2): Unit of the numerator  $(3.3 \times 10-5)$ , which seems cm K, should be added in the equation, because if the diameters are in nm in Equation (1), then the numerator would be  $(3.3 \times 102)$  nm K.

5. Page 8, line 7: The authors indicate that osmotic coefficients can be related to the square root of the molality by a 6th order polynomial function with considerable accuracy. How accurate, 1%? I would recommend to present the formula and give an example to demonstrate its accuracy.

6. Page 8, lines 8: Also for the osmotic coefficients, the authors mention that "it is diameter dependent and must be taken into account," but didn't clarify how to take the diameter-dependence into account.

7. Page 9, Table 1 and Figures 7-11: The diameter of dry particles is defined as "Xd" in the text, but in those table and figures, it is presented as "Xd". Please be consistent.

**ACPD**
8. Page 15, lines 26: The authors claim that "The sulfur in the fuel is oxidized to SO2, which then undergoes rapid oxidation to SO3 and subsequently to sulfuric acid..." I agree with the authors that all the fuel sulfur is oxidized to SO2, but disagree that oxidization from SO2 to SO3 is rapid. In fact, it is very inefficient ( $\sim$ 1-5%), as the two cited references indicated.

9. Page 16, line 8: Reference, Gysel et al. (2007), is not presented in the reference section. Please verify.

10. Page 16, line 14: For fuel sulfur content (FSC), the authors use the unit of  $\mu$ g of sulfur per g of fuel, but in Table 2, the authors also sue the unit of ppm. Please be consistent.

11. Page 16, line 14: I don't understand the meaning of "old and modern cruise conditions".

12. Could the authors provide an estimate of experimental uncertainties of the determined GF and SMF results in Section 5?

13. Page 21, line 5: the referenced journal should be "Atmos. Environ.".

**ACPD**

---

## Referee Comment (RC2) · Anonymous Referee #2 · 23 Aug 2018

Review of
**Application of a Hygroscopicity Tandem Differential Mobility Analyzer for characterizing PM Emissions in exhaust plumes from an Aircraft Engine burning Conventional and Alternative fuels**

By Max B. Trueblood et al.

The paper consists of two main parts: 1) quality control of a hygroscopicity tandem differential mobility analyser  (H-TDMA) and 2) experimental data in aircraft engine. The H-TDMA is designed for fast response and stable humidity conditions. The quality control is done with care and shows a well functioning system. The experimental data gives important information. There are, however, some things that are unclear and I recommend publication after major revision.

General comments and questions:

1        The authors define the hygroscopicity parameter *soluble mass fraction* (SMF).  This value depends on the knowledge or assumptions of the chemical character of the soluble material. In the literature, several parameters have been used for hygroscopicity. Kappa defined by Petters and Kreidenweis  (2007 and the discussion in ACPD) being the one mostly used today. But before that $\varepsilon$ representing the soluble fraction under assumptions of the chemical composition was used. $\varepsilon$ was abandoned, due to risks of misunderstanding.  Why are you choosing to use a new parameter, similar to the one earlier abandoned?

2        I have some comments on section 3 and the equations. a) Shouldn't it be $(X_w^3 - X_d^3)$ in eq. 1? It comes from the expression of the amount of water, I think. As far as I can see this has consequence for eq.8 and 9.
 b) Eq. 1 is only valid for SR close to 1 ( see e.g. Prupacher and Klett, page 173 in the 1997 edition). Have you analysed the errors made at the SR values relevant here (0.85-0.99 according to line 27 on page 4)
c) Even though the equations 2 and 3 are often expressed as they are here, I think it is unfortunate to give constants that actually have a unit, without expressing the unit. E.g. the constant in eq. 3 (4.3) includes the values of $\rho_w$ and $M_w$ as well as $\pi$. The value is 4.3 only if $\rho_w$ is given in kg/litre and $M_w$ in g/mole, and these are not the units in the SI system. The constant 1000 in eq. 7 comes from not using SI units for $M_s$. Also, the result of this is that eq 9 is given in terms of $\pi$ in some parts and with "combined" constants in other parts. As far as I can see, it would have been possible to simplify it, if the full expression of eq. 3 would have been used.

3        A more general comment to the calculations and the equations. Soot particles are in general agglomerates of many primary particles and their volume equivalent diameters are in general smaller than the mobility diameters. The particles can thus gain some secondary aerosol mass and water without increasing their mobility diameter. GF determined by H-TDMA systems can thus be below 1. I recommend that you take this into account or at least discuss it as a source of error.

4	Small changes in GF for values close to one can have large influence on the cloud forming ability of the particles. Can you specify for example the lowest GF that is significantly larger than 1? Or the uncertainty in small GFs in general?

5	P.8 l.31-32: It says: " the SR-calc for the largest two or three particle diameters was computed and an average was obtained". Does this mean that the theoretical curves in figure 2 are fitted to the largest sizes?

6	P.11 Fig.3 Are you sure that this is an effect of slow growth, and not of artefacts due to mismatch between DMA voltage and CPC counting? Have you tested with really slow voltage scans? Or scanning both up and down?

7	Are there any ways to control the SR or do you have to work with the SR you get? Why are you working with such a high SR? I think it would be good to motivate this in the paper.

8	You have chosen to work with an aerosol to flow ratio of 3/15.  A lower aerosol flow would increase the resolution and decrease the problem caused by a varying dN/dlogDp. Could you discuss this in more detail?

Details:

P.1 l. 31	Spell out all abbreviations, e.g. UHC.
P.3 l. 8-9	Have you tested if the charger is strong enough to neutralize the aerosol?
P.3 l.28-29	Have you made sure that the whole cooling volume is cooled equally effective and that there are no "pockets" of water that is not circulated?
P.4 l.9	Are the 104 increments equally separated on a linear or a logarithmic scale or separated in another way?
P.4 l.21	Performing HV2 sweeps on 12 different particle sizes in 9 minutes is very impressive!
P.4 l.27	The range in SR given is wide. Why is that?
P.6 Fig.1	Make sure that all symbols are defined. For example P1 and P2.
P.6 l.11	There is an extra "nm" in the beginning of the raw.
P.9 l. 13	Was the diameter 13,49 nm confirmed experimentally?
P.10 Fig.2	Consider the precision in the SR values. Also, please describe if the SR values are SR-calc or determined from the dew point sensor.
P.13 Fig.5	There seem to be a drop in SR over the period presented. Could you quantify this drop and expand the SR scale to make it more sensitive.
P.15 l.2	Why do you only present data from the 143 m location?
P.15 l.8-11	Did you see any bimodal GF distributions? I am especially thinking of the cases when the contribution from the engine was relatively small. Would it be possible to distinguish the engine particles from the ambient ones by there growth factors?
P.16 l.7	I guess you mean that the GF is close to 1 and not to 0? And in my opinion the GFs are not close to 1, they rather seem to be 1.2. What

could be the reason for this? It makes a large difference for their cloud forming ability.

P.16 l.11    Did you apply the factor 0.869 for all data, independent of the soluble fraction, that is also for particles that probably has no or very little sulphuric acid?

Fig. 7-11    Specify that the SMF assumes that the soluble material is sulphuric acid (if this is the case). Also for the low sulphur fuels.

The figures in general: Please provide error bars. The quality control should be able to result in error bars.

Reference:

Petters, M. D. and Kreidenweis, S. M.: A single parameter representation of hygroscopic growth and cloud condensation nucleus activity, Atmos. Chem. Phys., 7, 1961-1971, https://doi.org/10.5194/acp-7-1961-2007, 2007.

---

## Referee Comment (RC3) · R. Howard (Referee) · 31 Aug 2018

Journal: ACP

Title: Application of a Hygroscopicity-Tandem Differential Mobility Analyzer for characterizing PM Emissions in exhaust plumes from an Aircraft Engine burning Conventional and Alternative fuels

Author(s): Max B. Trueblood et al.
MS No.: acp-2018-507
MS Type: Research article

The authors describe well their analyzer system and pertinent application results. The aerosol hydroscopic property measurements are relevant, especially with the inclusion and comparison for alternative fuels. I highly recommend the paper for publication. I offer the following suggested edits and comments:

| Page # | Line # | Suggested Edits or Comments |
|---|---|---|
| 1 | 20 | Replace "fuels" with "types of fuel" |
| 1 | 23 | "decreased" is spelled incorrectly |
| 2 | 3 | Change "on" to "onto" |
| 2 | 16 | Place a comma after "plume" |
| 2 | 31 | Change "45s" to "45 s" or better,  change to "45 seconds" |
| 3 | 7 | "an ice bath to dry the sample" ... is not fully unclear. Please elaborate. |
| 3 | 14 | Change "poly-dispersed aerosol was classified" to "poly-dispersed aerosol was classified by size" |
| 3 | 16 | Throughout the paper, the tense switches from "was" to "is". I am not sure the change is always correct. Maybe an expert in grammar should review and make suggestions. |
| 3 | 17-18 | The word "now" is not necessary and should not be used in this manner in a formal paper. |
| 3 | 20 | Suggest that "Valves V2 and V3 can direct ..." be changed to "Valves V2 and V3 are used to ..." |
| 3 | 21 | Suggest that "Valves V4 and V5 achieve ..." be changed to "Valves V2 and V3 are used to achieve ..." |
| 3 | 25, 26, 28 | I consulted with colleagues and we are not familiar with the use of the term "thermostat" being used as a verb. Suggest not using the term "thermostated" in a formal paper. |
| 4 | 5 | Add comma after "frequency" |
| 4 | 6 | Change "it(1)" to "it (1)" |
| 4 | 31 | Add comma after "Thus" |
| 5 | 1 | The word "very" should not be used in a formal paper. |
| 6 | 2 | Suggest using a simpler Fig. 1 title with this long explanation in the body text. |
| 6 | 6 | "discuss" should be "discusses" |
| 6 | 9 | Change "deliver a given diameter Xd particle" to "deliver sample with a given diameter Xd particles" |
| 6 | 10 | "initiates voltage sweep" should be "initiates a voltage sweep" |
| 7 | 11-12 | Change "... this calibration is then later utilized ..." to " ... this calibration is utilized" |
| 7 | 7 | Change "when it is exposed ..." to "... when exposed ..." |
| 15 | 4 | Change "will be" to "are" |

| 15 | 22 | Change "20nm" to "20 nm" |
|---|---|---|
| 15 | 28 | Change "an existing soot particle to form a hybrid particle that subsequently has a significant water soluble component" to "existing soot particles to form hybrid particles that subsequently have significant water soluble components" |
| 16 | 15 | Change "This data is" to "These data are" [Throughout my career, my technical editing staff have required "data" to be treated as plural, regardless of the use. |
| 18 | 13 | Change "plots the GF" to "plots, the GF" or, better, combine with the previous sentence using "where" |
| 18 | 14 | Add comma, change "(FT plus THT) the GF is" to "(FT plus THT), the GF is" |
| 20 | 1 | Add colon, change "These findings are (1)" to " These findings are: (1)" |
| | | |

---

## Author Comment (AC1) · 18 Oct 2018

trueblud@mst.edu Received and published: 18 October 2018

Response to Reviews, Rev#1 Manuscript acp-2018-507

Anonymous Referee #1

We thank the referee for a very thorough review of our manuscript.

The referee's comments on various topics were very valuable and we believe that addressing these issues considerably improves the manuscript.

-reviewer's comments (in italic typeset, blue font). -a point-by-point response (in regular typeset, black font)

RC#1 There are a large number of abbreviations in this manuscript. I would recommend to include a list of abbreviations at the end of the text. RESPONSE #1: We have inserted a list of abbreviations at the end of the manuscript.

RC#2 Page 3, line 12: In the text, the authors use the unit "L m-1" as the abbreviation of liter per minute. Since "m" is also used as the abbreviation for meter in many places in this manuscript, I would recommend to change the unit of flow rate to "L min-1". RESPONSE #2: We have changed the units to "L min-1" in the manuscript.

RC#3 Page 7, line 9: The authors claim that RHOu and RHOs are assumed to be known, but they only provide the assumed value of RHOu based on a previously published study. What is the assumed value of RHOs used in this study? RESPONSE #3: These values are added in a table in the Supplemental Data.

RC#4 Equation (2): Unit of the numerator (3.3\_10-5), which seems cm K, should be added in the equation, because if the diameters are in nm in Equation (1), then the numerator would be (3.3\_102) nm K. RESPONSE #4: We have made this correction. . RC#5 Page 8, line 7: The authors indicate that osmotic coefficients can be related to the square root of the molality by a 6th order polynomial function with considerable accuracy. How accurate, 1%? I would recommend to present the formula and give an example to demonstrate its accuracy. RESPONSE #5: This has been added in the Supplemental Information. Both a table and formulas are presented.

RC#6 Page 8, lines 8: Also for the osmotic coefficients, the authors mention that "it is diameter dependent and must be taken into account," but didn't clarify how to take the diameter-dependence into account. RESPONSE #6: This is dealt with in Eq. (8) and in the Supplemental Information at the end.

RC#7 Page 9, Table 1 and Figures 7-11: The diameter of dry particles is defined as "Xd" in the text, but in those table and figures, it is presented as "Xd". Please be consistent. RESPONSE #7: That table has been deleted.

RC#8 Page 15, lines 26: The authors claim that "The sulfur in the fuel is oxidized to SO2, which then undergoes rapid oxidation to SO3 and subsequently to sulfuric acid: : :" I agree with the authors that all the fuel sulfur is oxidized to SO2, but disagree that oxidization from SO2 to SO3 is rapid. In fact, it is very inefficient (\_1-5%), as the two cited references indicated. RESPONSE #8: We have deleted the word "rapid".

RC#9 Page 16, line 8: Reference, Gysel et al. (2007), is not presented in the reference section. Please verify. RESPONSE #9: We have inserted this citation in the list of references.

RC#10 Page 16, line 14: For fuel sulfur content (FSC), the authors use the unit of \_g of sulfur per g of fuel, but in Table 2, the authors also sue the unit of ppm. Please be consistent. RESPONSE #10: We have corrected this, and use ppm throughout to be consistent.

RC#11 Page 16, line 14: I don't understand the meaning of "old and modern cruise conditions". RESPONSE #11: The text in the manuscript has been updated to refer to combustor inlet temperature conditions. The references to old and modern cruise conditions have been removed.

RC#12 Could the authors provide an estimate of experimental uncertainties of the determined GF and SMF results in Section 5? RESPONSE #12: We have removed the hygroscopic property SMF and replaced it with Kappa based on the recommendation of another reviewer. The experimental uncertainties in GF and Kappa are now provided in Section 5. We have included the following sentences in the revised manuscript: "The uncertainty in GF was 9% particles with diameter ~10 nm, and 3% for the larger diameters (26 nm). The uncertainty in  $\kappa$  was 7% and 2% for particles with diameter ~ 10 nm and ~26 nm, respectively."

СЗ

RC#13 Page 21, line 5: the referenced journal should be "Atmos. Environ.". RE-SPONSE #13: We have corrected this in the manuscript.

---

## Author Comment (AC2) · 18 Oct 2018

Response to Reviews, Rev#2 Manuscript acp-2018-507

Anonymous Referee #2

We thank the referee for a very thorough review of our manuscript.

The referee's comments on various topics were very valuable and we believe that addressing these issues considerably improves the manuscript.

–reviewer's comments (in italic typeset, blue font). –a point-by-point response (in regular typeset, black font)

The paper consists of two main parts: 1) quality control of a hygroscopicity tandem differential mobility analyser (H-Â■‐TDMA) and 2) experimental data on an aircraft engine. The H-‐TDMA is designed for fast response and stable humidity conditions. The quality control is done with care and shows a well functioning system. The experimental data gives important information. There are, however, some things that are unclear and I recommend publication after major revision.

General comments and questions:

RC#1 The authors define the hygroscopicity parameter soluble mass fraction (SMF). This value depends on the knowledge or assumptions of the chemical character of the soluble material. In the literature, several parameters have been used for hygroscopicity. Kappa defined by Petters and Kreidenweis (2007 and the discussion in ACPD) being the one mostly used today. But before that $\varepsilon$ representing the soluble fraction under assumptions of the chemical composition was used. $\varepsilon$ was abandoned, due to risks of misunderstanding. Why are you choosing to use a new parameter, similar to the one earlier abandoned? RESPONSE #1: That entire section has been deleted.

RC#2 I have some comments on section 3 and the equations. a) Shouldn't it be (Xw3-Â■‐Xd3) in eq. 1? It comes from the expression of the amount of water, I think. As far as I can see this has consequence for eq.8 and 9. RESPONSE #2A: No, it is correct as it stands. Please refer to Eq. (6-33), p146, Pruppacher and Klett, (1978).

Eq. 1 is only valid for SR close to 1 ( see e.g. Prupacher and Klett, page 173 in the 1997 edition). Have you analysed the errors made at the SR values relevant here (0.85-Â■‐0.99 according to line 27 on page 4) RESPONSE #2B: We have analyzed the error brought about by this and have discussed this in the paper.

Even though the equations 2 and 3 are often expressed as they are here, I think it is unfortunate to give constants that actually have a unit, without expressing the unit. E.g. the constant in eq. 3 (4.3) includes the values of w and Mw as well as $\pi$. The value is 4.3 only if w is given in kg/litre and Mw in g/mole, and these are not the units in the SI system. This has been corrected in the manuscript now.

The constant 1000 in eq. 7 comes from not using SI units for Ms. Also, the result of this is that eq 9 is given in terms of $\pi$ in some parts and with "combined" constants in other parts. As far as I can see, it would have been possible to simplify it, if the full expression of eq. 3 would have been used. RESPONSE #2: This has been corrected in the manuscript.

RC#3 A more general comment to the calculations and the equations. Soot particles are in general agglomerates of many primary particles and their volume equivalent diameters are in general smaller than the mobility diameters. The particles can thus gain some secondary aerosol mass and water without increasing their mobility diameter. GF determined by H-‐TDMA systems can thus be below 1. I recommend that you take this into account or at least discuss it as a source of error. RESPONSE #3: Gysel et al., (2007) stated that 15% of any given combustion particle is not soot. Thus we reduced the diameter Xd to 1/1.15=0.87 of what the DMA1 provided.

RC#4 Small changes in GF for values close to one can have large influence on the cloud forming ability of the particles. Can you specify for example the lowest GF that is significantly larger than 1? Or the uncertainty in small GFs in general? RESPONSE #4: We have included the following sentences in the revised manuscript: "The uncertainty in GF was 9% particles with diameter $\sim$10 nm, and 3% for the larger diameters (26 nm). The uncertainty in $\kappa$ was 7% and 2% for particles with diameter $\sim$ 10 nm and $\sim$26 nm, respectively."

RC#5 P.8 l.31-‐32: It says: " the SR-‐calc for the largest two or three particle diameters was computed and an average was obtained". Does this mean that

the theoretical curves in figure 2 are fitted to the largest sizes? RESPONSE #5: That is correct.

RC#6 P.11 Fig.3 Are you sure that this is an effect of slow growth, and not of artefacts due to mismatch between DMA voltage and CPC counting? Have you tested with really slow voltage scans? Or scanning both up and down? RESPONSE #6: Great care was taken to determine the lag time between when a voltage was imposed on the central rod of DMA2 and when particles selected by that voltage arrived at the CPC. Such a calibration took several days and we are confident of our result.

RC#7 Are there any ways to control the SR or do you have to work with the SR you get? Why are you working with such a high SR? I think it would be good to motivate this in the paper. RESPONSE #7: The MST H-TDMA was purposely designed to take samples that do not last long and to operate in environments where the ambient conditions (say temperature) may change significantly and abruptly. It makes no sense to try to do a humidigram on a sample that is only present for perhaps 60 sec. Thus we opted to design an instrument that used only one SR, but held that SR very constant.

Other values of SR might be obtained by mixing air from the HUM and very dry air for both the polydisperse and the sheath air, but that would require more controls than we wanted to do. That might be an interesting path to pursue later on. It would allow longer operating times before the wetting tubes dried out.

Thus, if one can only have one SR condition, we deemed it best to use the condition that was most easily obtained and was the most stable.

RC#8 You have chosen to work with an aerosol to flow ratio of 3/15. A lower aerosol flow would increase the resolution and decrease the problem caused by a varying dN/dlogDp. Could you discuss this in more detail? RESPONSE #8: A lower Qp and higher Qs2 would, indeed, provide higher resolution. However, a lower Qp would also decrease the concentration seen by the CPC. A higher Qs2 would also increase the resolution, but again it would decrease the concentration seen by the CPC. Generally

speaking, the higher the resolution, the lower is the concentration seen by the CPC. As mentioned in the text, the concentrations were already somewhat low sometimes, so we did not want to exacerbate that situation. Also, the wetting tubes dry out sooner for higher Qs2 flow rates. We felt that we had an optimal outcome here.

Details:

RC#9 P.1 l. 31 Spell out all abbreviations, e.g. UHC. RESPONSE #9: This has been done and a list of abbreviations have been provided at the end.

RC#10 P.3 l. 8-Â∎‐9 Have you tested if the charger is strong enough to neutralize the aerosol? RESPONSE #10: The bipolar charger is capable of housing 1 to 4 units of Po-210. Each unit has a strength of 500 $\mu$Ci. That is 500 to 2,000 $\mu$Ci. Po-210 is an Alpha emitter, which makes it considerably better at charging aerosols than a Beta emitter (Kr-85). This is because the specific ionization (number of ions created per centimeter of travel) of Alpha particles is much greater than that of Beta particles. Since the concentration of the aircraft engine particles was rather low by the time the plume reached the 143m sampling location, we are confident that the bipolar charger was strong enough.

RC#11 P.3 l.28-Â∎‐29 Have you made sure that the whole cooling volume is cooled equally effective and that there are no "pockets" of water that is not circulated? RESPONSE #11: The volume of the water bath surrounding DMA2 was approximately 12 L. The flow rate through the bath was approximately 5 L/min. Thus there was a complete water exchange every two minutes, a fairly short time. Furthermore, we point out that the SR was determined by a self-calibration using challenge aerosols of pure chemicals. So even if there were non circulated water pocket, the effective SR was determined.

RC#12 P.3 l.9 Are the 104 increments equally separated on a linear or a logarithmic scale or separated in another way? RESPONSE #12: The logarithm of the voltage vs. time is linear.

RC#13 P.4 l.21 Performing HV2 sweeps on 12 different particle sizes in 9 minutes is very impressive! RESPONSE #13: The MST H-TDMA was designed to study samples that were available for a short period of time (such as the plume from an aircraft landing or taking off) and/or a sample from a source that is very expensive to operate. Thus we tried to maximize the amount of data. Thank you for the compliment.

RC#14 P.4 l.27 The range in SR given is wide. Why is that? RESPONSE #14: When this instrument was first deployed in the field, it was not as well insulated as it is now. Thus the SR-calc from using challenge aerosols of pure chemicals was somewhat lower, say 0.85 to 0.91. Now that it is better insulated the SR-calc values are typically 0.97.

RC#15 P.6 Fig.1 Make sure that all symbols are defined. For example P1 and P2. RESPONSE #15: P1 and P2 are now defined within the schematic.

RC#16 P.6 l.11 There is an extra "nm" in the beginning of the raw. RESPONSE #16: This has been corrected.

RC#17 P.9 l. 13 Was the diameter 13,49 nm confirmed experimentally? RESPONSE #17: No, it was not. But since the original diameter is computed using trustworthy equipment and we corrected that with theory, the 13.49 nm should be trustworthy.

RC#18 P.9 Fig.2 Consider the precision in the SR values. Also, please describe if the SR values are SR-Â■‐calc or determined from the dew point sensor. RESPONSE #18: The Reviewer must be referring to Fig 4. This is the plot of SR-calc vs Elapsed Time over a 240 min span. The ordinate is SR-calc. The uncertainly in SR-calc was stated in the figure caption as 0.008.

RC#19 P.13 Fig.5 There seem to be a drop in SR over the period presented. Could you quantify this drop and expand the SR scale to make it more sensitive. RESPONSE #19: This drop seems to be about 6 parts in 986, or approximately 0.6 parts in 100, or approximately 0.5% in a four hour period.

RC#20 P.15 l.2 Why do you only present data from the 143 m location? RESPONSE #20: The H-TDMA system was in a small trailer located 143 m downstream of the engine exit. This distance was deemed necessary to allow time for the hybrid particles to form from the insoluble cores and vapors of soluble species, such as H2SO4. Data from the other locations has been previously published in the literature (Moore et al., 2015). Only data acquired at the 143 m location are presented and discussed here to investigate the hygroscopic properties of the evolving plume.

RC#21 P.15 l.8-â■‐11 Did you see any bimodal GF distributions? I am especially thinking of the cases when the contribution from the engine was relatively small. Would it be possible to distinguish the engine particles from the ambient ones by there growth factors? RESPONSE #21: We do not observe any bimodal GF distributions.

RC#22 P.15 l.7 I guess you mean that the GF is close to 1 and not to 0? And in my opinion the GFs are not close to 1, they rather seem to be 1.2. What could be the reason for this? It makes a large difference for their cloud forming ability. RESPONSE #22: It should be noted that the growth factors never go to zero, even for the FT fuel, but rather to about 1.15. This may well be the result of the fact that the insoluble core is porous. Thus even if there is not a spherical shell of H2SO4 around it, there is probably H2SO4 in the pores.

RC#23 P.16 l.11 Did you apply the factor 0.869 for all data, independent of the soluble fraction, that is also for particles that probably has no or very little sulphuric acid? RESPONSE #23: Yes, we did. That may well be the reason that even the FT fuel showed a GF of 1.15 or so. Also, see the response to RC#22.

RC#24 Fig. 7-â■‐11 Specify that the SMF assumes that the soluble material is sulphuric acid (if this is the case). Also for the low sulphur fuels. RESPONSE #24: All references to SMF have been removed from the manuscript and replaced with Kappa based on the reviewer comment.

RC#25 The figures in general: Please provide error bars. The quality control should

be able to result in error bars. RESPONSE #25: Adding error bars to the plots would make them cluttered. We have instead included the uncertainty in the GF and Kappa values in the text.

Reference:

Petters, M. D. and Kreidenweis, S. M.: A single parameter representation of hygroscopic growth and cloud condensation nucleus activity, Atmos. Chem. Phys., 7, 1961-Â■‐1971, https://doi.org/10.5194/acp-Â■‐7-Â■‐1961-Â■‐2007, 2007.
* * *

---

## Author Comment (AC3) · 18 Oct 2018

Response to Reviews, Rev#3 Manuscript acp-2018-507

Referee #3 (Dr Robert Howard)

The authors describe well their analyzer system and pertinent application results. The aerosol hydroscopic property measurements are relevant, especially with the inclusion

and comparison for alternative fuels. I highly recommend the paper for publication. I offer the following suggested edits and comments:

We thank the referee for a very thorough review of our manuscript.

–reviewer's comments (in italic typeset, blue font). –a point-by-point response (in regular typeset, black font)

RC#1 Page 1, Line 20 Replace "fuels" w "types of fuel" RESPONSE #1: We have made that replacement.

RC#2 Page 1, Line 23 "decreased" is spelled incorrectly RESPONSE #2: We have made that correction.

RC#3 Page 2, Line 3 Change "on" to "onto" RESPONSE #3: We have made that correction.

RC#4 Page 2, Line 16 Place a comma after "plume" RESPONSE #4: We have made that correction.

RC#5 Page 2, Line 31 Change "45s" to "45 s" or better, change to "45 seconds" RESPONSE #5: We have made the correction to 45 seconds.

RC#6 Page 3, Line 7 "an ice bath to dry the sample" ... is not fully unclear. Please elaborate. RESPONSE #6: We have made that clarification.

RC#7 Page 3, Line 14: Change "poly-dispersed aerosol was classified" to "poly-dispersed aerosol was classified by size" RESPONSE #7: We have made that change.

RC#8 Page 3, Line 16 Throughout the paper, the tense switches from "was" to "is". I am not sure the change is always correct. Maybe an expert in grammar should review and make suggestions. RESPONSE #8: We have made these corrections.

RC#9 Page 3, Line 17-18 The word "now" is not necessary and should not be used in this manner in a formal paper. RESPONSE #9: We have made these corrections.

RC#10 Page 3, Line 20 Suggest that "Valves V2 and V3 can direct ..." be changed to "Valves V2 and V3 are used to ..." RESPONSE #10: We have made these corrections.

RC#11 Page 3, Line 21 Suggest that "Valves V4 and V5 achieve..." be changed to "Valves V4 and V5 are used to achieve..." RESPONSE #11: We have made these corrections.

RC#12 Page 3, Line 25-28 I consulted with colleagues and we are not familiar with the use of the term "thermostat" being used as a verb. Suggest not using the term "thermostated" in a formal paper. RESPONSE #12: We have made appropriate changes.

RC#13 Page 4, Line 5 Add comma after "frequency" RESPONSE #13: We have made that correction.

RC#14 Page 4, Line 6 Change "it(1)" to "it (1)" RESPONSE #14: We have made that change.

RC#15 Page 4, Line 31 Add comma after "Thus" RESPONSE #15: We have made that change.

RC#16 Page 5, Line 1 The word "very" should not be used in a formal paper. RESPONSE #16: We have made that change.

RC#17 Page 6, Line 2 Suggest using a simpler Fig. 1 title with this long explanation in the body text. RESPONSE #17: We have made that change.

RC#18 Page 6, Line 6 "discuss" should be "discusses" RESPONSE #18: We have made that change.

RC#19 Page 6, Line 9 Change "deliver a given diameter Xd particle" to "deliver sample with a given diameter Xd particles" RESPONSE #19: We have made that change.

RC#20 Page 6, Line 10 "initiates voltage sweep" should be "initiates a voltage sweep" RESPONSE #20: We have made that change.

RC#21 Page 7, Line 11-12 Change "... this calibration is then later utilized ..." to " ... this calibration is utilized" RESPONSE #21: We have made that change.

RC#22 Page 7, Line 7 Change "when it is exposed ..." to "... when exposed ..." RESPONSE #22: We have made that change.

RC#23 Page 15, Line 4 Change "will be" to "are" RESPONSE #23: We have made that change.

RC#24 Page 15, Line 22Change "20nm" to "20 nm" RESPONSE #24: We have made that change.

RC#25 Page 15, Line 28 Change "an existing soot particle to form a hybrid particle that subsequently has a significant water soluble component" to "existing soot particles to form hybrid particles that subsequently have significant water soluble components" RESPONSE #25: We have made that change.

RC#26 Page 16, Line 15 Change "This data is" to "These data are" [Throughout my career, my technical editing staff have required "data" to be treated as plural, regardless of the use. RESPONSE #26: We have made that change.

RC#27 Page 18, Line 13 Change "plots the GF" to "plots, the GF" or, better, combine with the previous sentence using "where" RESPONSE #27: We have made appropriate changes.

RC#28 Page 18, Line 14 Add comma, change "(FT plus THT) the GF is" to "(FT plus THT), the GF is" RESPONSE #28: We have made appropriate changes.

RC#29 Page 20, Line 1Add colon, change "These findings are (1)" to " These findings are: (1)" RESPONSE #29: We have made appropriate changes.

---

## Referee Report (RR1)

Journal: ACP

Title: Application of a Hygroscopicity-Tandem Differential Mobility Analyzer for characterizing PM Emissions in exhaust plumes from an Aircraft Engine burning Conventional and Alternative fuels

Author(s): Max B. Trueblood et al.
MS No.: acp-2018-507
MS Type: Research article

The authors clarified technical questions by reviewers, corrected minor grammatical issues and overall improved the document. As stated on the first review, the authors describe well their analyzer system and pertinent application results. The aerosol hydroscopic property measurements are relevant, especially with the inclusion and comparison for alternative fuels. I still highly recommend the paper for publication. I offer the following suggested edits and comments:

| Page # | Line # | Suggested Edits or Comments |
|---|---|---|
| 2 | 24 | Delete "measured" |
| 3 | 14 | Delete "were" |
| 3 | 18 | You are missing something between the numbers "0.91  0.99" |
| 8 | 6 | Period missing at the end of the sentence |
| 10 | 5 | Delete "was" |
| 10 | 7 | Sentence should not end with a preposition |
| 10 | 12 | Change "data is" to "data are" |
| 10 | 14 | Was taught to generally avoid using "very" in formal papers, but ok with me personally. |
| 10 | 18 | Delete the extra period at the end of the sentence |
| 11 | 5 | Remove parentheses around the "3" |
| 11 | 5 | Consider rewording to "Fig. 3 shows plots of κ vs. Xd for the same four chemicals |
| 13 | 6 | Delete the period after the "4" |
| 13 | 7 | Suggest adding the word "value" to read " … highest SR value …" |
| 13 | 7 | Add comma after the work "extremes" |
| 14 | 9 | Sentence seems out of order. You state "Also shown … " before you introduce the figure |
| 16 | 15 | Should be plural "analyses … have" |
| 17 | 3 | Put a space between the value and the units … (in several places) |
| 17 | 7 | Add space between value and units: 143 m instead of 143m |
| 17 |  | Suggest adding the words here in yellow: "The exhaust was transported ==through the tube== to a small trailer …" |
| 17 |  | Consider stating all dimensions in meters, maybe put 2" in parentheses |
| 17 | 17 | Add a space between the values and their units |
| 17 | 18 | Add a space between the value and the units |
| 17 | 29 | Wording indicates all the SO2 goes to SO3 and H2SO4. I don't think this is true. If you agree, add a qualifier like " a portion of the SO2 …" |
| 18 | 14 | Separate "withthe" |
| 19 | 4 | Add a space between the value and the units |
| 21 | 4 | Suggest adding the word particle → "… particle sources …" |

---

## Author Response (AR2)

19 November 2018

Dear Prof. Ari Laaksonen:

RE: Manuscript #:  acp-2018-507

Enclosed for your consideration is the revised manuscript titled "**Application of a Hygroscopicity-Tandem Differential Mobility Analyzer for characterizing PM Emissions in exhaust plumes from an Aircraft Engine burning Conventional and Alternative fuels**" (acp-2018-507). We would like to thank the reviewers for their thorough review of our paper, and their suggestions and recommendations.

You can see that this is our second round of reviewer comments and revisions / responses.

The reviewer's comments were valuable and we have incorporated them into our manuscript.  We hope that you will find the revised manuscript suitable for publication in Atmospheric Chemistry and Physics.

Sincerely,

Max B. Trueblood

Center of Excellence for Aerospace Particulate Emission Reduction Research
104 Schrenk Hall, 400 W 11th St
Missouri University of Science and Technology
Rolla, MO  65401
trueblud@mst.edu

Reviewer comments:

The authors clarified technical questions by reviewers, corrected minor grammatical issues and overall improved the document. As stated on the first review, the authors describe well their analyzer system and pertinent application results. The aerosol hydroscopic property measurements are relevant, especially with the inclusion and comparison for alternative fuels. I still highly recommend the paper for publication. I offer the following suggested edits and comments:

| Page # | Line # | Suggested Edits or Comments |
|---|---|---|
| 2 | 24 | Delete "measured" |
| 3 | 14 | Delete "were" |
| 3 | 18 | You are missing something between the numbers "0.91 0.99" |
| 8 | 6 | Period missing at the end of the sentence |
| 10 | 5 | Delete "was" |
| 10 | 7 | Sentence should not end with a preposition |
| 10 | 12 | Change "data is" to "data are" |
| 10 | 14 | Was taught to generally avoid using "very" in formal papers, but ok with me personally. |
| 10 | 18 | Delete the extra period at the end of the sentence |
| 11 | 5 | Remove parentheses around the "3" |
| 11 | 5 | Consider rewording to "Fig. 3 shows plots of $\kappa$ vs. Xd for the same four chemicals |
| 13 | 6 | Delete the period after the "4" |
| 13 | 7 | Suggest adding the word "value" to read " ... highest SR value ..." |
| 13 | 7 | Add comma after the work "extremes" |
| 14 | 9 | Sentence seems out of order. You state "Also shown ... " before you introduce the figure |
| 16 | 15 | Should be plural "analyses ... have" |
| 17 | 3 | Put a space between the value and the units ... (in several places) |
| 17 | 7 | Add space between value and units: 143 m instead of 143m |
| 17 | | Suggest adding the words here in yellow: "The exhaust was transported through the tube to a small trailer ..." |
| 17 | | Consider stating all dimensions in meters, maybe put 2" in parentheses |
| 17 | 17 | Add a space between the values and their units |
| 17 | 18 | Add a space between the value and the units |
| 17 | 29 | Wording indicates all the SO2 goes to SO3 and H2SO4. I don't think this is true. If you agree, add a qualifier like " a portion of the SO2 ..." |
| 18 | 14 | Separate "withthe" |
| 19 | 4 | Add a space between the value and the units |
| 21 | 4 | Suggest adding the word particle → "... particle sources ..." |

Response:

*We thank the reviewer for his/her valuable comments. We have made all the changes as the reviewer suggested above.*